# Effects of multi-scale heterogeneity on the simulated evolution of ice-rich permafrost lowlands under a warming climate

Jan Nitzbon[1,2,3], Moritz Langer[1,2], Léo C. P. Martin[3], Sebastian Westermann[3], Thomas Schneider von Deimling[1,2], and Julia Boike[1,2]

[1]Permafrost Research, Alfred Wegener Institute Helmholtz Centre for Polar and Marine Research, Potsdam, Germany
[2]Geography Department, Humboldt-Universität zu Berlin, Berlin, Germany
[3]Department of Geosciences, University of Oslo, Oslo, Norway

**Correspondence:** Jan Nitzbon (jan.nitzbon@awi.de)

**Abstract.**

In continuous permafrost lowlands, thawing of ice-rich deposits and melting of massive ground ice leads to abrupt landscape changes called thermokarst, which have widespread consequences on the thermal, hydrological and biogeochemical state of the subsurface. However, macro-scale land surface models (LSMs) do not resolve such localized subgrid-scale processes and could hence miss key feedback mechanisms and complexities which affect permafrost degradation and the potential liberation of soil organic carbon in high latitudes. Here, we extend the CryoGrid 3 permafrost model with a multi-scale tiling scheme which represents the spatial heterogeneities of surface and subsurface conditions in ice-rich permafrost lowlands. We conducted numerical simulations using stylized model setups to assess how different representations of micro- and meso-scale heterogeneities affect landscape evolution pathways and the amount of permafrost degradation in response to climate warming. At the micro-scale, the terrain was assumed to be either homogeneous or composed of ice-wedge polygons, and at the meso-scale to be either homogeneous or resembling a low-gradient slope. We found that by using different model setups and parameter sets, a multitude of landscape evolution pathways could be simulated which correspond well to observed thermokarst landscape dynamics across the Arctic. These pathways include the formation, growth, and gradual drainage of thaw lakes, the transition from low-centred to high centred ice-wedge polygons, and the formation of landscape-wide drainage systems due to melting of ice wedges. Moreover, we identified several feedback mechanisms due to lateral transport processes which either stabilize or destabilize the thermokarst terrain. The amount of permafrost degradation in response to climate warming was found to depend primarily on the prevailing hydrological conditions which in turn are crucially affected by whether or not micro- and/or meso-scale heterogeneities were considered in the model setup. Our results suggest that the multi-scale tiling scheme allows to simulate ice-rich permafrost landscape dynamics in a more realistic way than simplistic one-dimensional models, and thus facilitates more robust assessments of permafrost degradation pathways in response to climate warming. Our modelling work improves the understanding of how micro- and meso-scale processes affect the evolution of ice-rich permafrost landscapes, and it informs macro-scale modellers focusing on high-latitude land surface processes about the necessities and possibilities for the inclusion of subgrid-scale processes such as thermokarst within their frameworks.

# 1 Introduction

Thawing of permafrost in response to climatic change poses a threat to ecosystems, infrastructure, and indigenous communities in the Arctic (AMAP, 2017; Vincent et al., 2017; Schuur and Mack, 2018; Hjort et al., 2018). Pan-Arctic modelling studies have suggested substantial permafrost loss (Lawrence et al., 2012; Slater and Lawrence, 2013) and associated changes in the water and carbon balance of the permafrost region (Burke et al., 2017; Kleinen and Brovkin, 2018; Andresen et al., 2020) within the course of the twenty-first century and beyond. The potential liberation of carbon dioxide and methane from thawing permafrost poses a positive feedback to global climate warming (Schuur et al., 2015; Schneider von Deimling et al., 2015) which is yet poorly constrained. Indeed, the macro-scale[1] models used to project the response of permafrost to climate warming employ a simplistic representation of permafrost thaw dynamics, which only reflects gradual and spatially homogeneous top-down thawing of frozen ground. In particular, such models lack representation of thaw processes in ice-rich permafrost deposits that cause localized and rapid landscape changes called thermokarst (Kokelj and Jorgenson, 2013; Olefeldt et al., 2016). Thermokarst activity is induced on small spatial scales that are below the grid resolution of macro-scale land surface models (LSMs), but it can have widespread effects on the ground thermal and hydrological regimes (Fortier et al., 2007; Liljedahl et al., 2016), on soil erosion (Godin et al., 2014), and on carbon decomposition pathways (Lara et al., 2015; Walter Anthony et al., 2018; Turetsky et al., 2020). Thermokarst activity can induce positive feedback processes leading to accelerated permafrost degradation and landscape collapse (Turetsky et al., 2019; Farquharson et al., 2019; Nitzbon et al., 2020), but also stabilizing feedbacks have been observed in ice-rich terrain (Jorgenson et al., 2015; Kanevskiy et al., 2017). Overall, thermokarst processes can be considered a key factor of uncertainty in future projections of how the energy, water, and carbon balances of permafrost environments will respond to Arctic climate change (Turetsky et al., 2019).

Ice-rich permafrost landscapes prone to thermokarst activity are typically characterized by marked spatial heterogeneities that can be linked to the accumulation and melting of excess ice (Kokelj and Jorgenson, 2013). For example, polygonal-patterned tundra in the continuous permafrost zone is underlain by networks of massive ice wedges which give rise to a regular and periodic partitioning of both the surface and the subsurface at the micro-scale (Lachenbruch, 1962). At the meso-scale, past thermokarst activity has led to a high abundance of thaw lakes in ice-rich permafrost lowlands (Morgenstern et al., 2011). Excess ice melt on the micro-scale can lead to the emergence of thaw features and feedbacks on the meso-scale, such as the formation of drainage networks (Liljedahl et al., 2016), the lateral expansion of thaw lakes (Jones et al., 2011), or the development of thermo-erosional gullies (Godin et al., 2014). These features have the potential to interact with each other, thereby adding more complexity to the landscape evolution, for example, when a thaw lake drains upon incision of the thermo-erosional gully (Morgenstern et al., 2013). Overall, there is emerging evidence from field observations (Jorgenson et al., 2006; Farquharson et al., 2019) and remote sensing (Liljedahl et al., 2016; Nitze et al., 2018, 2020) that these subgrid-scale processes crucially affect permafrost degradation pathways and hence also the potential liberation of frozen carbon pools.

Numerical models which represent the *thermokarst-inducing processes* (Nitzbon et al., 2020) that give rise to the observed complex pathways of permafrost landscape evolution are an important tool to improve our understanding of how subgrid-scale

---

[1] See Section 2.1 and Table 1 for the terminology used in this manuscript to designate different spatial scales.

processes affect permafrost degradation in response to climate warming (Rowland et al., 2010; Turetsky et al., 2019). In recent years, substantial progress has been made in the development of numerical models to study the thermal and hydrological dynamics of permafrost terrain on small sptial scales (Painter et al., 2016; Jafarov et al., 2018), and to identify important feedbacks associated with various thermokarst landforms such as ice-wedge polygons (Abolt et al., 2018; Nitzbon et al., 2019; Abolt et al., 2020) or peat plateaus (Martin et al., 2019). The development of numerical schemes simulating ground subsidence resulting from excess ice melt (Lee et al., 2014; Westermann et al., 2016) enabled the assessment of transient changes of thermokarst terrain using dedicated permafrost models (Langer et al., 2016; Nitzbon et al., 2019), but also more broadly using the frameworks of LSMs (Lee et al., 2014; Aas et al., 2019).

Several of these models employed a so-called "tiling approach" to account for subgrid-scale heterogeneities of permafrost terrain. Instead of discretizing extensive landscape domains on a high-resolution mesh, the landscape is partitioned into a low number of characteristic landscape units, each of which is associated with a representative "tile" in the model. Thereby, geometrical characteristics of the landscape units (e.g., size and shape, symmetries, adjacencies) are used to parametrize lateral fluxes among the tiles. For example, Langer et al. (2016) used a two-tile model setup to investigate the effect of lateral heat fluxes in a lake-rich permafrost landscape, Nitzbon et al. (2019) suggested a three-tile setup to represent the micro-scale heterogeneity associated with ice-wedge polygon tundra, and Schneider von Deimling et al. (2020) applied a five-tile setup to represent the interaction of linear infrastructure such as roads with underlying and surrounding permafrost. So far, the tiling approach has not been applied to simultaneously represent heterogeneities of permafrost landscapes and their interactions across multiple spatial scales.

The overall scope of the present study was to investigate the effect of micro- and meso-scale heterogeneities on the transient evolution of ice-rich permafrost lowlands under a warming climate. Specifically, we addressed the following objectives:

1. To identify degradation pathways and feedback processes associated with lateral fluxes of mass and energy on the micro- and meso-scale.

2. To quantify permafrost degradation in terms of thaw-depth increase and ground subsidence in dependence of the representation of micro- and meso-scale heterogeneities.

For this, we introduced a multi-scale tiling scheme into the CryoGrid 3 permafrost model and conducted numerical simulations for a site in northeast Siberia under a strong twenty-first century climate warming scenario (Representative Concentration Pathway (RCP) 8.5). We considered different model setups reflecting either the micro-scale heterogeneity associated with ice-wedge polygons, the meso-scale heterogeneity associated with low-gradient slopes, or a combination of both. As a reference, we considered "single-tile" simulations which emulate the behaviour of macro-scale LSMs. Overall, our goal was to provide a scalable framework for exploring the transient evolution of permafrost landscapes in response to a warming climate which could potentially be incorporated into LSMs to allow more robust projections of permafrost loss under climate change. The presented simulations should thus be considered as numerical experiments to identify the spatial scales, environmental factors, and feedback mechanisms which affect the degradation of ice-rich permafrost.

## 2 Methods

### 2.1 Terminology for spatial scales

Throughout this manuscript we used a consistent terminology to refer to different characteristic length scales of landscape features and processes. This terminology is summarized in Table 1.

### 2.2 Model description

#### 2.2.1 Multi-scale tiling to represent spatial heterogeneity

We used the concept of laterally coupled tiles (Langer et al., 2016; Aas et al., 2019; Nitzbon et al., 2019) to represent subgrid-scale spatial heterogeneities of permafrost terrain. In general, the tiling concept involves the partitioning of real-world landscapes into a certain number of characteristic units, which are associated with the major surface (or subsurface) heterogeneities found in the landscape. Each of these units is then represented by a single "tile" in a permafrost model, and multiple tiles

 can interact through lateral exchange processes. The tiling approach thus allows to simulate subgrid-scale heterogeneities and lateral fluxes in macro-scale models like LSMs without discretizing extensive landscape domains on a high-resolution mesh, and hence keeping computational costs at a reasonable level.

For the present study, we applied the tiling concept at multiple scales to represent common spatial heterogeneities of ice-rich continuous permafrost lowlands (see Figure 1). At the micro-scale, these landscapes are typically characterized by ice-wedge

 polygons which give rise to a regular patterning of the landscape (see Appendix A for an example site from northeast Siberia). Here, we adopted the three-tile approach by Nitzbon et al. (2019), which partitions polygonal tundra lowlands into polygon centres, polygon rims and inter-polygonal troughs. At the meso-scale, tundra lowlands are characterized by gently-sloped terrain and often feature abundant thaw lakes that formed in the past due to thermokarst activity (see Appendix A). Here, we used three meso-scale tiles to represent a low-gradient slope which is efficiently drained at its lowest elevation ("drainage

 point").

Figure 2 provides an overview of the four model setups which were investigated in this study. These setups differ with respect to the number of micro-scale tiles ($N^\mu$) and meso-scale tiles ($N^\mathrm{m}$), the product of which amounts to the total number of tiles ($N = N^\mu \cdot N^\mathrm{m}$):

a. *Single-tile* ($N^\mu = 1$, $N^\mathrm{m} = 1$, Figure 2 a): This is the most simple case which reflects homogeneous surface and subsurface conditions across all spatial scales via only one tile (H). Water can drain laterally into an external reservoir at a fixed

 elevation ($e_\mathrm{res}$). This setup emulates the one-dimensional representation of permafrost in LSMs and corresponds to the setup used for the simulations conducted by Westermann et al. (2016).

b. *Polygon* ($N^\mu = 3$, $N^\mathrm{m} = 1$, Figure 2 b): This setting reflects the micro-scale heterogeneity associated with ice-wedge polygons via three tiles: polygon centers (C), polygon rims (R) and inter-polygonal troughs (T). The heterogeneity of the

 surface topography is expressed in different initial elevations of the soil surface of the tiles. The subsurface stratigraphies

of the tiles differ with respect to the depth ($d_x$) and amount ($\theta_x$) of excess ice, reflecting the subsurface ice-wedge network which is linked to the polygonal pattern at the surface. Lateral fluxes of heat, water, snow, and sediment are enabled among the tiles, and the trough tile is connected to an external water reservoir. This model setup has previously been used by Nitzbon et al. (2019) and Nitzbon et al. (2020).

c. *Low-gradient slope* ($N^\mu = 1$, $N^m = 3$, Figure 2 c): This setup reflects a meso-scale gradient of slope $S^m$, for which the outer (i.e., downstream) tile (H$^o$) is well-drained into an external reservoir. The intermediate (H$^m$) and inner (H$^i$) tiles represent the landscape upstream of the outer tile at constant distances ($D^m$). The setting assumes a translational symmetry perpendicular to the direction of the slope, i.e., each tile represents the same areal proportion at the meso-scale. Lateral surface and subsurface water fluxes are enabled among the meso-scale tiles, while lateral heat fluxes as treated
by Langer et al. (2016) were not considered at this scale.

    d. *Low-gradient polygon slope* ($N^\mu = 3$, $N^m = 3$, Figure 2 d): This setup reflects a meso-scale slope featuring ice-wedge polygons at the micro-scale. Each meso-scale tile includes three micro-scale tiles (C,R,T), corresponding to the *polygon* setup described above. However, only the trough tile of the outer polygon (T$^o$) is connected to an external reservoir (well-drained), while the intermediate and inner trough tiles are hydrologically connected along the meso-scale slope.

In the initial configurations, the landscapes were assumed to be undegraded, i.e., the ice-wedge polygons were low-centred (LCPs), and no thaw lakes (water bodies, WBs) were present along the meso-scale slope. However, the model allowed thermokarst features like high-centred polygons (HCPs) and thaw lakes to develop dynamically as a consequence of excess ice melt at different scales.

### 2.2.2   Processes representations (CryoGrid 3)

Each of the tiles described in section 2.2.1 was associated with a one-dimensional representation of the subsurface through the CryoGrid 3 permafrost model, which is a physical process-based land surface model tailored for applications in permafrost environments (Westermann et al., 2016).

    *Heat diffusion with phase change:* The numerical model computes the subsurface temperatures ($T(z,t)$ [°C]) by solving the heat diffusion equation, thereby taking into account the phase change of soil water ($\theta_w$ [-]) through an effective heat capacity
($C_{eff}(z,T)$):

$$\underbrace{\left( C(z,T) + \rho_w L_{sl} \frac{\partial \theta_w}{\partial T} \right)}_{C_{eff}(z,T)} \frac{\partial T}{\partial t} = \frac{\partial}{\partial z}\left( k(z,T) \frac{\partial T}{\partial z} \right) \tag{1}$$

In Eqn. (1) $k(z,T)$ [W K$^{-1}$ m$^{-1}$] denotes the thermal conductivity and $C(z,T)$ [J K$^{-1}$ m$^{-3}$] the volumetric heat capacity of the soil, both parametrized depending on the soil constituents. $\rho_w$ [kg m$^{-3}$] is the density of water, and $L_{sl}$ [J kg$^{-1}$ K$^{-1}$] the specific latent heat of fusion of water. The upper boundary condition to Eq. (1) is prescribed as a ground heat flux ($Q_g$ [W m$^{-2}$]) which
is obtained by solving the surface energy balance as described in Westermann et al. (2016). The lower boundary condition is given by a constant geothermal heat flux at the lower end of the model domain ($Q_{geo} = 0.05$ W m$^{-2}$).

*Snow scheme:* CryoGrid 3 simulates the dynamic build-up and ablation of a snow pack above the surface, heat conduction through the snowpack, changes to the snowpack due to infiltration and refreezing of rain and meltwater, and changes in snow albedo due to ageing. Snow is deposited at an initial density ($\rho_{snow} = 250\,\mathrm{kg\,m^{-3}}$) which can increase due to infiltration and refreezing of water.

*Hydrology scheme:* The model further employs a simple vertical hydrology scheme to represent changes in the ground hydrological regime due to infiltration of rain or meltwater, and evapotranspiration (Martin et al., 2019; Nitzbon et al., 2019). Infiltrating water is instantaneously routed downwards through unfrozen soil layers, whose water content is set equal to the field capacity parameter (i.e., the water holding capacity): $\theta_{fc} = 0.5$. Once a frozen soil layer is reached, excess water successively saturates the above-lying unfrozen soil layers upwards. Excess water is allowed to pond above the surface, leading to the formation of a surface water body. Surface water is modulated by evaporation as well as lateral fluxes to adjacent tiles or into an external reservoir (see paragraph *Lateral fluxes* below). Heat transfer through unfrozen surface water bodies is realized by assuming complete mixing of the water column during the ice-free period (i.e., the water column has a constant temperature profile with depth).

*Excess ice scheme:* CryoGrid 3 has an excess ice scheme ("Xice"), which enables it to simulate the ground subsidence as a result of excess ice melt (i.e., thermokarst), following the algorithm proposed by Lee et al. (2014): Subsurface grid cells which have an ice content ($\theta_i$) that exceeds the natural porosity ($\phi_{nat}$) of the soil constituents ($\theta_m + \theta_o$) are treated as excess-ice-bearing cells; once such a cell thaws, the resulting excess water is routed upwards, while above-lying soil constituents are routed downwards and fill the space previously occupied by the excess ice. According to this scheme, thawing of a excess-ice-bearing grid cell of thickness $\Delta d$ results in a net ground subsidence ($\Delta s$) which equals

$$\Delta s = \Delta d \underbrace{\frac{\theta_i - \phi_{nat}}{1 - \phi_{nat}}}_{\theta_x} , \tag{2}$$

where $\theta_x$ denotes the excess ice fraction of the ice-rich soil cell (see Appendix B for a derivation). The excess water is then treated by the hydrology scheme, i.e., it can either pond at the surface or run off laterally.

*Lateral fluxes:* The thermal regime and thaw processes in ice-rich permafrost are affected by lateral fluxes of mass and energy at subgrid-scales. We used the concept of laterally coupled tiles to represent subgrid-scale heterogeneities of permafrost terrain (see Section 2.2.1 for details). We followed Nitzbon et al. (2019) to represent lateral fluxes of heat, water, and snow between adjacent tiles at the micro-scale. Furthermore, we included micro-scale advective sediment transport due to slumping, following the approach and using the same parameter values as Nitzbon et al. (2020). At the meso-scale, we only considered lateral water fluxes, for which we further discriminated between surface and subsurface contributions. Both surface and subsurface water fluxes were calculated according to a gradient in water table elevations following Darcy's law, but the hydraulic conductivities differed considerably ($K^{subs} = 10^{-5}\,\mathrm{m\,s^{-1}}$, $K^{surf} = 10^{-2}\,\mathrm{m\,s^{-1}}$; see Appendix C for details). We did not consider lateral fluxes of heat, snow, and sediment at the meso-scale, as these were assumed to be either negligible on the time scale of interest (heat, sediment), or of uncertain importance (snow). Finally, the model allows for drainage of water into an "external reservoir" at a

fixed elevation ($e_{\text{res}}$) and a constant hydraulic conductivity ($K_{\text{res}} = 2\pi K_{\text{subs}}$; see Supplementary Information of Nitzbon et al. (2020) for details).

## 2.3 Model settings and simulations

### 2.3.1 Study area

While the the scientific objectives and the modelling concept pursued in this study are general and in principle transferable to any ice-rich permafrost terrain, we chose Samoylov Island in the Lena River delta in northeast Siberia as our focus study area. The island lies in the continuous permafrost zone and is characterized by ice-wedge polygons and surface water bodies of different sizes. The model input data (soil stratigraphies, parameters, meteorological forcing, etc.) were specified based on available observations from this study site. Details on the study area are provided in Appendix A.

### 2.3.2 Soil stratigraphies and ground ice distribution

The subsurface composition of all tiles was represented via a generic soil stratigraphy (Table 2). The stratigraphy was based on previous studies that applied CryoGrid 3 to the same study area (Nitzbon et al., 2019, 2020). It consists of two highly porous layers of $0.1\,\text{m}$ thickness which reflect the surface vegetation and an organic-rich peat layer. Below those, a mineral layer with silty texture follows. An excess ice layer of variable total ice content $\theta_{\text{i}}$ extends from a variable depth $d_{\text{x}}$ down to a depth of $10.0\,\text{m}$. Between the variable excess ice layer and the mineral layer, we assumed an ice-rich intermediate layer of $0.2\,\text{m}$ thickness and a total ice content of $0.65$. Below the variable excess ice layer follow an ice-poor layer and bedrock down to the end of the model domain. The default ice content assumed for the excess ice layer of homogeneous tiles (no micro-scale heterogeneity) was $\theta_{\text{i}}^{\text{H}} = 0.75$ and the default depth of this layer was $d_{\text{x}} = 0.9\,\text{m}$. To reflect the heterogeneous excess ice distribution associated with ice-wedge polygons, different ice contents were assumed for the excess ice layers of polygon centres ($\theta_{\text{i}}^{\text{C}} = 0.65$), rims ($\theta_{\text{i}}^{\text{R}} = 0.75$), and troughs ($\theta_{\text{i}}^{\text{T}} = 0.95$). However, the area-weighted mean ice content of ice-wedge polygon terrain is identical to the default value assumed for homogeneous tiles (Table 3).

### 2.3.3 Topography and geometrical relations among the tiles

*Topography:* For the *single-tile* setup (Figure 2 a) the initial elevation of the soil surface was set to $0.0\,\text{m}$. This value does not affect the simulation results, but it serves as a reference for the variation of the micro- and meso-scale topographies. For the *polygon* setup (Figure 2 b) we assumed that the rims were elevated by $e_{\text{R}} = 0.2\,\text{m}$ relative to the polygon centres, while the troughs had the same initial elevation as the centres ($e_{\text{T}} = 0.0\,\text{m}$). While this choice of parameters varies slightly from the values assumed in previous studies (Nitzbon et al., 2019, 2020), it allows for consistent comparability to the setups without micro-scale heterogeneity (see Table 3). The meso-scale topography in the *low-gradient slope* setup (Figure 2 c) was obtained by multiplying the meso-scale distances ($D^{\text{m}}$) with the slope of the terrain ($S^{\text{m}}$). The micro-scale topography of the *low-gradient polygon slope* setup (Figure 2 d) was obtained by summing up the relative topographic elevations of the meso- and micro-scales.

*Geometry:* We made simplifying assumptions to determine the adjacency and geometrical relations such as distances and contact lengths among the tiles at the micro- and meso-scale. These geometrical characteristics determine the magnitude of lateral fluxes between the tiles, and only need to be specified if more than one tile is used to represent the respective scale. For the *polygon* setup with three micro-scale tiles, we assumed a geometry of a circle (centre tile) surrounded by rings (rim and trough tiles), identical to the setup by Nitzbon et al. (2020). This geometry is fully defined by specifying the total area ($A^\mu$) of a single polygonal structure together with the areal fractions ($\gamma_{\mathrm{C,R,T}}$) of the three tiles. Here, we chose values which constitute a compromise between observations from the study area and comparability to the setups without micro-scale heterogeneity (Table 3). For the employed geometry, the micro-scale distances ($D^\mu$) and contact lengths ($L^\mu$) can be calculated according to the formulas provided in the Supplementary Information to Nitzbon et al. (2020). For the *low-gradient slope* setups with three meso-scale tiles, we assumed translational symmetry of the landscape in the direction perpendicular to the direction of the gradient. Furthermore, the three meso-scale tiles were assumed to be at equal distances of $D^{\mathrm{m}} = 100\,\mathrm{m}$ from each other. Hence, each meso-scale tile is representative of the same areal fraction of the overall landscape.

### 2.3.4   Meteorological forcing data

We used the same meteorological forcing dataset which has been used in preceding studies based on CryoGrid 3 for the same study area (Westermann et al., 2016; Langer et al., 2016; Nitzbon et al., 2020). The dataset spans the period from 1901 until 2100 and is based on down-scaled CRU-NCEP v5.3 data for the period until 2014. For the period after 2014, climatic anomalies obtained from CCSM4 projections for the RCP8.5 scenarios were applied to a fifteen-year climatological base period (2000-2014). A detailed description of how the forcing dataset was generated is provided by Westermann et al. (2016).

### 2.3.5   Simulations

Table 4 provides an overview of all simulations which were conducted for the four different model setups introduced in section 2.2.1. For both the *single-tile* and the *polygon* setups we conducted four model runs in which we varied the elevation of the external water reservoir ($e_{\mathrm{res}}$) in order to reflect a broad range of hydrological conditions. For the *low-gradient slope* and the *low-gradient polygon slope* setups we conducted two model runs in which we varied the gradient of the slope ($S^{\mathrm{m}}$) in order to reflect essentially flat ($S^{\mathrm{m}} = 0.001$) as well as gently-sloped terrain ($S^{\mathrm{m}} = 0.01$). In all *slope* simulations, one tile (H$^{\mathrm{o}}$ and T$^{\mathrm{o}}$, respectively) was connected to an external reservoir (see Fig. 2 c,d; $e_{\mathrm{res}} = -10.0\,\mathrm{m}$).

The subsurface temperatures of each model run were initialized in Oct 1999 with a typical temperature profile for that time of year which was based on long-term borehole measurements from the study area (Boike et al., 2019). Using multi-year spin-up periods did not result in significant changes to the near-surface processes investigated in this study so that biases related to the initial subsurface temperature profile can be excluded. The analyzed simulation period was the twenty-first century from Jan 2000 until Dec 2099.

## 3 Results

The presentation of the results is structured according to the four different model setups described in Section 2.2.1 and visualized in Figure 2 a-d. Figures 3 to 6 provide an overview of the simulated landscape evolution in the four setups by displaying the landscape configuration in selected years during the simulation period. While these figures are intended to allow an intuitive overview of the qualitative behaviour of the model, Figures 7 and 8 provide a quantitative assessment of permafrost degradation by showing time series of the maximum thaw depths and accumulated ground subsidence for all simulations.

### 3.1 Single-tile

Figure 3 shows the landscape configuration for selected years throughout the twenty-first century under RCP8.5 for the *single-tile* setup (see Figure 2 a) under four different hydrological boundary conditions. Until the middle of the simulation period (2050) no excess ice melt and associated ground subsidence occur, irrespective of the hydrological conditions (Fig. 3 a–c, all columns; Fig. 8 a). However, the maximum thaw depths are steadily increasing during that period (see Figure 7 a), with negligible differences between the four runs under different hydrological conditions. Between 2050 and 2075 excess ice melt occurs in all *single-tile* simulations, leading to ground subsidence of about $0.2\,\mathrm{m}$ in the two well-drained cases (Figure 3 d), first and second column), about $0.1\,\mathrm{m}$ in the intermediate case (Figure 3 d, third column), and about $0.5\,\mathrm{m}$ in the poorly-drained case, where excess ice melt leads to the formation of a shallow surface water body (Figure 3 d, fourth column). By 2100 total ground subsidence amounts to about $0.8\,\mathrm{m}$ in the well-drained simulations (Fig. 4 e, first and second column), and about $1.0\,\mathrm{m}$ in the simulation with $e_{\mathrm{res}} = -0.5\,\mathrm{m}$, where it is accompanied by the formation of a shallow surface water body (Fig. 4 e, third column). In the poorly-drained simulation ($e_{\mathrm{res}} = -0.1\,\mathrm{m}$), excess ice melt proceeds fastest, causing the surface water body to deepen, and to reach a depth of about $2\,\mathrm{m}$ by 2100; a talik of about about $1.5\,\mathrm{m}$ thickness has formed underneath the water body by the end of the simulation period (Fig. 3 e, fifth column).

Overall, the simulation results indicate that permafrost degradation is strongest as soon as a limitation of water drainage results in the formation of a surface water body. The presence of surface water changes the energy transfer at the surface in different ways. First, it reduces the surface albedo (from $0.20$ for barren ground to $0.07$ for water), resulting in a higher portion of incoming shortwave radiation. Second, water bodies have a high heat capacity which slows down their freeze-back compared to soil. Third, the thawed saturated deposits beneath the surface water body have a higher thermal conductivity compared to unsaturated deposits, which allows heat to be transported more efficiently from the surface into deeper soil layers.

Interestingly, during the initial phase of excess ice melt which occurs between 2050 and 2075, our simulations suggest a non-monotonous dependence of permafrost degradation on the drainage conditions. This is indicated by the fact that the lowest degradation both in terms of maximum thaw depth and accumulated subsidence is simulated for the intermediate case with $e_{\mathrm{res}} = -0.5\,\mathrm{m}$ (see Figure 7 a and 8 a). This can likely be attributed to contrasting effects of the hydrological regime in the active layer on thaw depths. When the near-surface ground is unsaturated (as in the simulations with $e_{\mathrm{res}} = -1.0\,\mathrm{m}$ and $e_{\mathrm{res}} = -10.0\,\mathrm{m}$), the highly-porous organic-rich surface layers (see Table 2) have an insulating effect on the ground below due their low thermal conductivity. At the same time, less heat is required to melt the ice contained in the mineral soil layers

whose ice content corresponds to the field capacity than it would be needed if their pore space was saturated with ice. In the intermediate case with $e_{\text{res}} = -0.5\,\text{m}$, the combination of dry, insulating near-surface layers and ice-saturated mineral layers beneath leads to the lowest thaw depths and hence the slowest initial permafrost degradation. However, as soon as a surface water body forms in that simulation (between 2075 and 2100), the positive feedback on thaw described above takes over, resulting in stronger degradation by 2100 compared to the well-drained settings ($e_{\text{res}} = -1.0\,\text{m}$ and $e_{\text{res}} = -10.0\,\text{m}$) for which no surface water body forms during the simulation period.

In summary, the *single-tile* simulations illustrate the non-trivial relation between the ground hydrological regime and permafrost thaw, and demonstrate the positive feedback associated with surface water body formation resulting from excess ice melt.

## 3.2  Polygon

Figure 4 illustrates the landscape evolution throughout the twenty-first century under RCP8.5 for the simulations with the *polygon* setup, i.e., with a representation of micro-scale heterogeneity typical for ice-wedge polygonal tundra. In addition, it shows the geomorphological state of the polygon micro-topography, i.e., whether it is low-centered (LCP), intermediate-centered (ICP), high-centred with inundated troughs (HCPi), high-centred with drained troughs (HCPd), or covered by a surface water body (WB), according to Equations (D1) to (D5) in Appendix D. Initially, all simulations feature undegraded ice-wedge polygons with a low-centred micro-topography (Figure 4 a).

Between 2000 and 2025 a shallow surface water body of about $0.5\,\text{m}$ depth forms in the poorly-drained simulation ($e_{\text{res}} = 0.0\,\text{m}$) as a result of excess ice melt in the center, rim, and trough tiles (Fig. 4 b, fourth column). The bottom of the water body has a high-centred topography. The landscape configuration does not change much until 2050 (Fig. 4 c, fourth column), and the maximum thaw depths and accumulated ground subsidence increase more slowly than in the initial two to three decades (Figures 7 b and 8 b, purple lines). We explain this interim stabilization with the subaqueous transport of sediment from the centres to the rims and from the rims to the troughs, where the additional sediment has an insulating effect on the ice wedge underneath. After 2050 excess ice melt proceeds faster, causing the surface water body to deepen, reaching a depth between about $1\,\text{m}$ (centre tile) and more than $2\,\text{m}$ (trough tile) by 2075 (Fig. 4 d, fourth column). We explain the acceleration of permafrost degradation at the beginning of the second half of the simulation period (Figures 7 b and 8 b, purple lines) by a combination of additional warming from the meteorological forcing, and positive feedbacks due to the surface water body (as explained for the *single-tile* simulation in Section 3.1). Until 2100 excess ice melt proceeds further, resulting in a water body depth of 2-4 m and the formation of an extended talik underneath (Fig. 4 e, fifth column). By the end of the simulation period, the lake is not entirely bottom-freezing in winter. This constitutes another positive feedback on the degradation rate, since the heat from the ground can be transported much more efficiently through bottom-freezing water bodies than through those which do not bottom-freeze, due to the higher thermal conductivity of ice compared to that of unfrozen water.

The remaining three simulations show a similar landscape evolution during the first half of the simulation period (2000-2050), with excess ice melt occurring only in the trough tiles, resulting in a transition from LCP to ICP micro-topography (Fig. 4 a–c, first to third column). In contrast to the *single-tile* setup, the *polygon* simulations show a monotonous relation between

permafrost degradation and drainage conditions during this period. Higher elevations of the external water reservoir (i.e., poorer drainage) result in faster thaw-depth increase (Figures 7 b) and earlier onset of ground subsidence (Figure 8 b). Between 2050 and 2075, substantial excess ice melt occurs in the trough and rim tiles, involving a transition from the ICP to a pronounced HCP micro-topography (Fig. 4 d, first to third column). In addition to the positive feedback through surface water formation, the degradation rate in the *polygon* simulations is accelerated by a positive feedback through lateral snow redistribution. Snow is deposited preferentially in micro-topographic depressions such as initially subsided troughs where it improves the insulation and traps heat in the subsurface during winter, enabling faster and deeper thaw in the subsequent summer. Note that this positive feedback is not represented in the *single-tile* simulations.

By 2075 the influence of the drainage conditions on the surface water coverage is most pronounced. While a water body extending over all tiles has formed in the poorly-drained simulation ($e_{\mathrm{res}} = 0.0\,\mathrm{m}$), the polygon centres are (still) elevated above the water table in the intermediate case with $e_{\mathrm{res}} = -0.5\,\mathrm{m}$. For $e_{\mathrm{res}} = -1.0\,\mathrm{m}$, surface water is found only in the trough tile, while all surface water is drained in the simulation with $e_{\mathrm{res}} = -10.0\,\mathrm{m}$. Irrespective of the hydrological boundary conditions, permafrost degradation continues in the final decades until 2100. While the high-centred polygon in the well-drained simulation ($e_{\mathrm{res}} = -10.0\,\mathrm{m}$) remains free of surface water (Fig. 4 e, first column), surface inundation increases in the runs with intermediate hydrological conditions (Fig. 4 e, second and third column), resulting in the formation of a water body of 1-3 m depth underlain by a talik in the simulation with $e_{\mathrm{res}} = -0.5\,\mathrm{m}$. However, the water body is (still) bottom-freezing in 2100, in contrast to the water body with floating ice in the simulation with $e_{\mathbf{res}} = 0.0\,\mathrm{m}$.

Overall, the *polygon* simulations reveal a marked dependence of the pathways of landscape evolution and associated permafrost degradation on the hydrological conditions, consistent with previous results under recent climatic conditions (Nitzbon et al., 2019).

### 3.3 Low-gradient slope

While the *single-tile* and *polygon* simulations illustrated general feedbacks due to excess ice melt and micro-scale lateral fluxes, they cannot reflect heterogeneous landscape dynamics at the meso-scale. The simulations with the *low-gradient slope* setup illustrate how meso-scale lateral water fluxes in gently-sloped terrain give rise to diverse pathways of landscape evolution and how these depend on the slope gradient ($S^{\mathrm{m}}$).

Figure 5 shows the landscape evolution throughout the twenty-first century for the simulations with the *low-gradient slope* setup, i.e., with a meso-scale representation according to a slope but without micro-scale heterogeneities. Irrespective of the slope gradient, the simulated evolution of the outer tiles is very similar to that of the well-drained *single-tile* simulations ($e_{\mathrm{res}} = -10.0\,\mathrm{m}$ and $e_{\mathrm{res}} = -1.0\,\mathrm{m}$) throughout the entire simulation period (compare Figure 3 first and second columns with Figure 5 first and fourth column). The outer tiles are stable until about the half of the simulation period. During the second half, excess ice melt sets in and the ground subsides at an increasing rate, reaching an accumulated subsidence of about 0.8 m by 2100. The similarity to the well-drained *single-tile* simulations can be explained by the fact that the outer tile is very efficiently drained ($e_{\mathrm{res}} = -10.0$), such that the lateral water input from the intermediate tile is directly routed further into the external reservoir. Hence, the "upstream" influence on the outer tile becomes negligible.

For both slope gradients, the evolution of the intermediate tile is similar to that of the inner tile. However, the evolution of the two tiles is different among the two simulations for different slope gradients. For the lower slope gradient ($S^{\mathrm{m}} = 0.001$), which reflects an essentially flat landscape, melting of excess ice and associated ground subsidence occur during the first half of the simulation period, and a shallow layer of surface water (about $0.2\,\mathrm{m}$) forms in these tiles (Fig. 5 a–c, second and third column). By 2050, the soil surface in the intermediate and inner tiles has subsided below the surface elevation of the outer tile such that the initial slope is dissipated. After 2050 excess ice melt proceeds faster and the surface water body in the intermediate and inner tiles reaches a depth of almost $1\,\mathrm{m}$ by 2075 (Fig. 5 d, second and third column). The permafrost table in these tiles lowered by about $2\,\mathrm{m}$ relative to its initial position. Between 2075 and 2100 a talik forms beneath the water body in the two inland tiles (Fig. 5 e, second and third column) which by 2100 reaches a depth of about $2.0\,\mathrm{m}$. The ground subsidence in the outer tile during the second half of the simulation enables surface and subsurface water transport from the intermediate tile into the external reservoir, and thus leads to a lowering of the water level of the water body in the intermediate and inner tiles (Fig. 5 d–e). By 2100 the water body depth is about $1.2 - 1.5\,\mathrm{m}$, which is significantly lower than the water body of about $2.0\,\mathrm{m}$ depth which forms in the poorly-drained *single-tile* simulations (Fig. 3 e, fourth column). This difference in water body depths is despite excess ice melt and surface water formation set on several decades earlier in the *low-gradient slope* simulation than in the poorly-drained *single-tile* simulation. These different pathways of water body and talik formation illustrate that there is no linear relationship between the water body depth and talik extent, since lateral interactions at the meso-scale can give rise to different transient dynamics.

For the higher slope gradient ($S^{\mathrm{m}} = 0.01$), which reflects moderately-sloped terrain, initial excess ice melt occurs in the intermediate and inner tiles between 2000 and 2025, and by 2050 these have subsided by about $0.2\,\mathrm{m}$ (Fig. 5, fifth and sixth column), similar to the simulation with $S^{\mathrm{m}} = 0.001$. By 2075, ground subsidence has reached about $0.5\,\mathrm{m}$ in the intermediate and inner tiles. The overall shape of the slope does not change much because excess ice melt occurs also in the outer tile during that time. By 2100, the intermediate and inner tile have subsided by about $1.2\,\mathrm{m}$, and the outer tile by about $0.8\,\mathrm{m}$, resulting in a slightly concave slope. The presence of a thin layer of surface water in the intermediate and inner tiles in the years 2025, 2075, and 2100 indicates a wetter hydrological regime compared to the outer tile. However, as the moderate slope is sustained throughout the simulation, it facilitates drainage of surface water and precludes the formation of a surface water body as it is the case in the simulation with the flatter topography ($S^{\mathrm{m}} = 0.001$). We suspect, however that if the simulations would be prolonged beyond 2100, the excess ice melt in the intermediate and inner tiles would likely continue at a faster rate than in the outer tile and this would result in the reversal of the initial slope and promote the formation of an inland surface water body.

While the overall dynamics of the three-tile slope simulations could also be reflected using a two-tile setup, we would like to point to the small but significant differences between the evolution of the intermediate tile and the inner tile. For both slope gradients, the intermediate tile shows a slightly stronger degradation rate than the inner tile. This can be explained due to the additional water input from the inner tile which sustains saturated conditions in the near-surface soil layers during the thawing season. The inner tile in turn lacks this lateral water input and is thus more likely to develop dry, thermally insulating conditions in the near-surface soil.

Overall, the slope gradient ($S^{\mathrm{m}}$) has a similar influence on permafrost degradation in the *low-gradient slope* setup, as the elevation of the external reservoir ($e_{\mathrm{res}}$) has in the *single-tile* simulations (see Figures 7 a,c and 8 a,c). This can be attributed to the direct influence of the slope gradient on drainage efficiency. Due to the positive feedbacks related to the formation of a surface water body, the overall permafrost degradation in the setting with $S^{\mathrm{m}} = 0.001$ is almost twice as much as in the setting with $S^{\mathrm{m}} = 0.01$ for which no water body forms. We note that the maximum thawed ground and the accumulated ground

subsidence by 2100 in both *low-gradient slope* simulations are within those simulated for the two extreme settings in the *single-tile* simulations.

### 3.4    Low-gradient polygon slope

Finally, we consider the most complex setup which reflects gently-sloped polygonal tundra via nine tiles, incorporating heterogeneities on both the micro- and the meso-scale. Figure 6 shows different stages of the simulated landscape evolution in the

390 *low-gradient polygon slope* setup throughout the twenty-first century for two different slope gradients $S^{\mathrm{m}}$.

     Like the outer tiles in the *low-gradient slope* setup, the outer polygons show an evolution which is very similar to that of the well-drained (single) *polygon* simulation with $e_{\mathrm{res}} = -10.0\,\mathrm{m}$, irrespective of the slope gradient (compare Fig. 6 first and fourth column with Fig. 4 first column): until 2050 initial excess ice melt occurs in the trough tiles, entailing a transition from LCP to ICP topography; during the second half of the simulation period, permafrost degradation proceeds at an increasing

rate and brings about a transition to a pronounced HCP topography with the troughs having subsided by about $2\,\mathrm{m}$. Again, the similarity between the outer polygon evolution and the well-drained single polygon evolution can be explained by the efficient drainage of the outer polygon troughs which leads to a negligible "upstream" influence.

     For the lower slope gradient ($S^{\mathrm{m}} = 0.001$), the intermediate and inner polygons show an evolution which is similar to each other, but different from each of the single *polygon* simulations (compare Fig. 4 with Fig. 6 second and third column). Between

400 2000 and 2025 excess ice melt in the rim and trough tiles leads to ponding of surface water in the intermediate and inner polygons (Fig. 6 b, second and third column) with the soil surface of the polygon centre being close to or elevated above the water table. Besides a lowering of the water table, the configuration of these polygons does not change much until 2050 (Fig. 6 c, second and third column) when both have developed an HCPi topography, i.e., inundated rims and troughs which subsided below the level of the centres. Between 2050 and 2075, excess ice melt continues, leading to a pronounced high-centered

topography (Fig. 6 d, second and third column). Meanwhile, the surface water disappears from the rim and trough tiles, as a consequence of marked excess ice melt in the outer polygons rim and troughs tiles during that period (Fig. 6 d, first column). The soil surface elevation of the trough tiles increases from the outer towards the inner polygon, allowing for efficient drainage of the entire landscape. Between 2075 and 2100 all three polygons show a similar evolution, resulting in pronounced high-centered topographies with troughs about $2\,\mathrm{m}$ deep and rims that subsided by about $1\,\mathrm{m}$ in total. Note that between 2075 and

2100 the troughs of the intermediate and inner polygons subsided more than that of the outer polygon, resulting in an decreased drainage efficiency. Hence, surface water starts to pond again in the intermediate and inner troughs (Fig. 6 e, second and third column). Overall, the *low-gradient polygon slope* simulation with $S^{\mathrm{m}} = 0.001$ shows a transient phase in which the formation

of a water body is initiated, but its growth is impeded due to the formation of efficient drainage pathways through the subsiding troughs in the outer ("downstream") polygon.

The simulated landscape evolution for the moderate slope gradient ($S^m = 0.01$) differs from that for the lower slope gradient during the first half of the simulation period. Until 2025, excess ice melt is restricted to the trough tiles, leading to a transition from LCP to ICP topography in all three polygons (Fig. 6 b, fifth and sixth column). In the intermediate and inner polygons, permafrost degradation proceeds faster than in the outer polygon, such that these develop a HCPd topography by 2050. The faster degradation can be explained by the overall wetter hydrological regime compared to the very efficiently drained outer polygon. However, the meso-scale slope still allows for sufficient drainage of surface water, such that (in contrast to the simulation with $S^m = 0.001$) the ponding of surface water is mostly precluded (Fig. 6 c, fifth and sixth column). During the second half of the simulation period, the evolution of the polygons is very similar for both slope gradients. Ice-wedge degradation proceeds at an increasing pace between 2050 and 2100, resulting in polygons with pronounced HCP topography. While the troughs of the outer polygons are efficiently drained (HCPd, Fig. 6 e, first and fourth column), some surface water pools in the troughs of the intermediate and inner polygons (HCPi, Fig. 6 second, third, fifth and sixth column).

The overall similarity among the two *low-gradient polygon slope* simulations for different slope gradients is also reflected in the time series of maximum thaw depths (Fig. 7 d) and accumulated ground subsidence (Fig. 8 d). During a transient phase spanning roughly from 2010 until 2070, the permafrost degradation in the simulation with $S^m = 0.001$ is slightly higher than in the simulation with $S^m = 0.01$. Before and after this transient period, the maximum thaw depths and the accumulated ground subsidence are almost identical for the two settings. This is a qualitative difference to the *low-gradient slope* simulations without micro-scale heterogeneity, where the simulation with $S^m = 0.001$ resulted in almost twice the amount of permafrost degradation compared to the simulation with $S^m = 0.01$ (Fig. 7 c,d and Fig. 8 c,d). The projected permafrost degradation in both *low-gradient polygon slope* simulations is confined within the range spanned by the projections for the *polygon* setup (see Fig. 7 b, d and Fig. 8 b, d). However, by 2100 the projected maximum thaw depths and the accumulated ground subsidence are very close to the single *polygon* simulation under well-drained conditions ($e_{res} = -10.0\,\mathrm{m}$).

While the *low-gradient polygon slope* setup provides the most complex representation of landscape heterogeneities – and hence potentially captures most feedback processes – the projected pathways of landscape evolution and permafrost degradation showed little sensitivity to the initial gradient of the meso-scale slope, and, overall, the projected landscape response was more gradual than in the other setups which feature abrupt initiation or acceleration of permafrost degradation (Figures 7 and 8). We also note that the *low-gradient polygon slope* setup is the only one, for which the formation of a talik was not projected for any of the tiles and for any of the parameter settings (Fig. 6 a–e).

## 4 Discussion

### 4.1 Simulating permafrost landscape degradation pathways under consideration of micro- and meso-scale heterogeneities

The multi-scale tiling approach introduced in this study (see Section 2.2.1) facilitates the simulation of degradation pathways of ice-rich permafrost landscapes in response to a warming climate, under consideration of feedbacks emerging from heterogeneities and lateral fluxes on subgrid-scales. The following qualitative assessment based on observed landscape changes across the Arctic demonstrates that the model is able to realistically represent the dominant pathways of landscape evolution and important feedback processes induced by permafrost thaw and excess ice melt.

The most idealized cases considered were *single-tile* simulations without heterogeneities on the micro- or meso-scale, corresponding to previous simulations with models that incorporate a representation of excess ice melt and ground subsidence (Lee et al., 2014; Westermann et al., 2016). The simulations with this setup reflect the fundamental modes of landscape change due to excess ice melt (i.e., thermokarst) (Kokelj and Jorgenson, 2013). These changes are ranging from the gradual subsidence of a well-drained "upland" setting to the formation of thaw lakes in a water-logged "lowland" setting. The marked sensitivity of the simulations to the prescribed hydrological conditions underlines that landscape evolution and permafrost degradation crucially depend on whether water from melted excess ice drains or ponds at the surface. On the one hand, the simulations reveal a positive feedback on permafrost degradation through ponding of water from melted excess ice at the surface. The presence of surface water alters the energy exchange with the atmosphere and increases the heat input into the ground (albedo, thermal properties), which in turn allows for additional excess ice melt. This positive feedback is in agreement with previous observations (Connon et al., 2018; O'Neill et al., 2020) and modelling results (Rowland et al., 2011; Langer et al., 2016; Westermann et al., 2016). On the other hand, simulations under well-drained conditions favoured stabilization due to the improved insulation by dry organic surface layers which is also in agreement with observations (Göckede et al., 2017, 2019) and modelling results (Martin et al., 2019; Nitzbon et al., 2019).

In the *polygon* setup, micro-scale heterogeneities associated with ice-wedge polygons and represented lateral fluxes of heat, water, snow, and sediment were considered. The simulated landscape evolution pathways under a warming climate correspond to those simulated and discussed by Nitzbon et al. (2020) (for the "Holocene Deposits" stratigraphy). They range from the rapid formation of a deep surface water body to the development of high-centred polygons with a pronounced relief. While observations of initial ice-wedge degradation and the development of high-centred polygons are reported across the Arctic (Farquharson et al., 2019; Liljedahl et al., 2016), the likelihood of thaw lake formation in ice-wedge terrain is debated and likely depends on the overall wedge-ice content (Kanevskiy et al., 2017). Despite the same ground ice distribution in terms of excess ice content ($\theta_x$) and depth ($d_x$) on average (see Table 3), the simulated permafrost degradation in the *polygon* settings exceeds that in the respective *single-tile* simulations, suggesting that positive feedbacks (e.g., preferential accumulation of snow and water in troughs) exceed the influence of stabilizing feedbacks (e.g., slumping of sediment from rims into troughs). Hence, the simulations suggest that the presence of micro-scale heterogeneities can crucially affect the timing and the rate of permafrost degradation in ice-rich lowlands. Ultimately, both the *single-tile* and the *polygon* setups are limited by the

prescription of static hydrological boundary conditions which do not allow transient changes between different pathways but rather prescribe the possible landscape evolution a priori.

In the *low-gradient slope* setup, the hydrological conditions in the inland part of the model domain can develop dynamically and the water level in a given part of the landscape adapts to topographic changes in adjacent parts due to the representation of meso-scale lateral water fluxes. From an almost flat initial topography ($S^m = 0.001$, Fig. 5 left), the formation of a thaw lake in the inland has been simulated, succeeded by the gradual drainage of the lake in response to ground subsidence in the outer part. This setup thus captures in a stylized way the competing mechanisms of thaw lake formation and growth on the one hand, and (gradual) lake drainage on the other hand. Gradual or partial drainage of mature thaw lakes has been observed (Morgenstern et al., 2013) and simulated (Kessler et al., 2012), and we note that also the thermokarst lakes in the southeastern and southwestern part of Samoylov Island are indicative of gradual drainage (Figure A1 a). This gradual lake drainage constitutes a potential negative feedback on permafrost degradation, since shallower water bodies are more likely to bottom-freeze, allowing more efficient cooling of the ground during wintertime (Boike et al., 2015; Langer et al., 2016; O'Neill et al., 2020). Previous modelling studies have also demonstrated that the stability and the thermal regime of permafrost in the vicinity of thaw lakes is affected by meso-scale lateral heat fluxes from taliks forming underneath the lakes (Rowland et al., 2011; Langer et al., 2016). These effects have not been considered in this study, but constitute another feedback due to lateral fluxes at the meso-scale.

Finally, the most complex pathways of landscape evolution are revealed, when both micro- and meso-scale heterogeneities are taken into account. The simulation of a *low-gradient polygon slope* reflects the transition from low-centred polygonal tundra with inundated centres and troughs towards efficiently drained terrain with a high-centred polygon relief. This transient landscape evolution which resembles well the schematic evolution of polygonal tundra depicted by (Liljedahl et al., 2016) has not been simulated in a numerical model before, as it involves an interaction between micro-scale (i.e., ice-wedge degradation) and meso-scale (i.e., drainage along slope) processes. Due to the inclusion of a wide range of feedback processes on multiple scales, we consider the simulations conducted with the *low-gradient polygon slope* setup to most realistically mirror the real-world dynamics of ice-rich permafrost lowlands. Interestingly, the projected thaw-depth increase and ground subsidence showed little sensitivity to the meso-scale slope graident (Fig. 7 d and 8 d), suggesting a higher robustness of these projections against parameter variations compared to the more simple setups.

Despite the capacity of reflecting a wide range landscape evolution pathways, certain forms of landscape change that have been observed in ice-rich permafrost landscapes are not possible to reflect using the presented model framework due to a lack of necessary process parametrizations. For instance, real-world thaw lakes are known to drain "catastrophically" instead of gradually upon the incision of a lake basin by a thermo-erosional gully. This can happen as a result of either gully growth or lake expansion (Jones et al., 2011; Kessler et al., 2012), both of which are a consequence of lateral erosion. This process is, however, not represented in our model at the meso-scale. Similarly, the dynamic development of retrogressive thaw slumps and their interactions with other landforms would require additional parametrizations of mass-wasting processes as well as slope- and aspect-dependent modifications of incoming radiation (Lewkowicz, 1986). Beyond this, it should be noted that further potential feedbacks mechanisms are not represented in our model, such as lateral heat and sediment transport at the meso-

scale, or ecological processes such as vegetation succession, all of which potentially alter the ground thermal and hydrological regimes (Shur and Jorgenson, 2007; Kokelj and Jorgenson, 2013; Kanevskiy et al., 2017) and affect permafrost thaw.

## 4.2 Implications for simulating high-latitude land surface processes using macro-scale models

The presence of permafrost in high-latitude landscapes crucially affects hydrological and biogeochemical processes and makes their realistic representation within macro-scale LSM frameworks challenging (Chadburn et al., 2017; Andresen et al., 2020; Burke et al., 2020). Beyond the requirement to accurately simulate freeze-thaw processes near the surface, the presence of excess ice in the subsurface is an additional source of complexity. Indeed, its melting involves rapid changes in the topography which affect hydrological, ecological, and biogeochemical processes, and these in turn feed back on the thermal dynamics of the subsurface. Here, our results support previous findings which have suggested that micro- (Abolt et al., 2018, 2020; Nitzbon et al., 2019) and meso-scale (Langer et al., 2016; Jafarov et al., 2018) heterogeneities and lateral fluxes exert important controls on the simulated thermal, hydrological, and biogeochemical state of the subsurface in ice-rich terrain. Beyond the importance of subgrid-scale heterogeneity, our simulations suggest that interactions *across* scales can introduce further complexities and feedbacks in the land surface evolution in response to climate warming, which are not represented in LSMs. For example, the rapid subsidence of inter-polygon troughs due to melting of ice wedges can entail landscape-wide regime shifts of the soil moisture state and runoff generation. Similarly, the setups which reflect micro- and/or meso-scale heterogeneity enable to simulate pathways of landscape evolution in which some parts of the landscape are inundated (e.g., deepening troughs), while other parts are subject to drained conditions. The spatial variability of the ground thermal and hydrological regimes directly affects other ecosystem processes such as the decomposition of soil organic carbon (Lara et al., 2015). In fact, several studies have suggested that small water bodies could constitute "hotspots" for carbon decomposition and greenhouse gas emissions (Laurion et al., 2010; Abnizova et al., 2012; Langer et al., 2015; Elder et al., 2020), which would be missed in projections of macro-scale models used to constrain the global permafrost carbon-climate feedback (Schuur et al., 2015).

The modelling approach of multi-scale landscape tiling pursued in our study constitutes a promising way to represent subgrid-scale heterogeneities of permafrost landscapes without the need to increase the spatial resolution of a macro-scale model. Our modelling suggests that key feedback processes acting at and across the micro- and meso-scales of ice-rich terrain could be represented in a nine-tile setup, corresponding to an increase in computational demand by about an order of magnitude. Despite mathematical and technical challenges that would need to be addressed before multi-scale tiling schemes could be incorporated within established LSMs (Fisher and Koven, 2020), we consider the concept of multi-scale tiling to bear considerable potential for a computationally affordable representation of subgrid-heterogeneity and lateral interactions in high-latitude ecosystems within LSM frameworks.

## 4.3 Towards simulating the cyclic evolution of ice wedges and thaw lakes in thermokarst terrain

Beyond the discussed implications for simulating permafrost degradation in response to the expected climate warming within the course of the twenty-first century, our modelling work reveals novel potentials for investigating the geomorphological evolution of thermokarst terrain under both past and future climatic conditions. It has been suggested that thermokarst terrain

is evolving in a cyclic manner on centennial-to-millennial timescales. For example, cycles of degradation and stabilization have been suggested to describe the evolution of ice wedges (Jorgenson et al., 2006, 2015; Kanevskiy et al., 2017) on these time scales. Similarly, but at a larger spatial scale, the cyclic evolution of thaw lakes has been hypothesized (Billings and Peterson, 1980) (see Grosse et al. (2013) for a review of the hypothesis). While we do not want to discuss here to which extent ice-rich permafrost landscapes have been evolving or are evolving according to these models, we would like to mention that the model framework presented in this study is able to capture a wide range of the processes involved in these cycles. Figure 9 provides a schematic of the cyclic evolution of thermokarst terrain and highlights the capacities of our model to simulate different pathways of (cyclic) landscape evolution.

The initial and advanced degradation of ice wedges and the associated transition from low-centered to high-centered polygons was captured by the model setup by Nitzbon et al. (2019). Nitzbon et al. (2020) complemented this by the stabilization of ice wedges due to sediment accumulation in the troughs, and also showed that it is possible to simulate ice-wedge collapse and thaw lake formation in response to climate warming scenarios. The simulations with representation of landscape heterogeneity and lateral water fluxes at the meso-scale, which are presented in this study, allow to capture further feedbacks. First, the formation of efficient drainage pathways through degrading ice-wedge networks, as depicted by Liljedahl et al. (2016), was qualitatively captured by the simulation using the *low-gradient polygon slope* setup (Fig. 2 d and 6). While the simulated landscape drainage did not completely interrupt the ice-wedge degradation (as indicated in Fig. 9), it did lead to a significantly slower degradation compared to the simulations without meso-scale heterogeneity (Fig. 7 b,d). Second, representing meso-scale lateral water fluxes (Fig. 2 c,d) enables our model in principle to simulate the drainage of thaw lakes, which would result in the exposure of unfrozen ground (talik) to the atmosphere. While it is not possible to simulate the catastrophic drainage of lakes as discussed above, subsidence of the terrain surrounding the thaw lake, can lead to its gradual drainage on longer timescales than considered here. It is further likely that using a model setup with a more complex meso-scale landscape topology, could result in more rapid drainage of water bodies.

As indicated in Fig. 9, several processes are not represented in our model, such that it cannot capture the full cycles of ice-rich permafrost evolution. In order to represent the initial stabilization of ice wedges more realistically, further ecological processes like vegetation succession and organic matter formation would need to be represented (Jorgenson et al., 2015; Kanevskiy et al., 2017). However, lateral sediment transport into the troughs captures this initial stabilization stage to a certain extent. The advanced stabilization of ice wedges involves the formation of ice-rich layers consisting of segregation ice above the massive ice wedge. The process of ice-segregation would require a more sophisticated subsurface hydrology scheme, representing the migration of liquid pore water towards the freezing front, as well as vertical displacement of sediment resulting from frost heave (Ballantyne, 2018). Finally – and probably most importantly – the formation of wedge ice is not yet represented in the model, which is necessary to form the initial stage of undegraded ice wedges from various other evolutionary stages (Fig. 9). In contrast to the formation of segregation ice, wedge-ice formation involves both vertical and horizontal processes, which presupposes the formation of frost cracks due to mechanical stress, which in turn is controlled by climatic conditions at the ground surface and at the top of permafrost, and mechanical properties of the soil (Lachenbruch, 1962; Matsuoka et al., 2018). Despite the lack of parametrizations for these processes, our numerical model developments provide important progress

towards an improved understanding of the long-term geomorphological evolution of periglacial landscapes under both, past and future, climatic conditions.

## 5   Conclusions

In the present study, we employed a multi-scale tiling approach in the CryoGrid 3 numerical permafrost model to assess the response of ice-rich permafrost lowlands to a warming climate. Specifically, we explored the sensitivity of the simulated pathways of landscape evolution and permafrost degradation to different representations of micro- and meso-scale heterogeneities. From the results of this study, we draw the following conclusions:

1. Representing micro- and meso-scale heterogeneities facilitates the simulation of manifold pathways of permafrost landscape evolution such as the dynamic formation and drainage of thaw lakes, the transition from low-centred to high-centred ice-wedge polygons, and the formation of meso-scale drainage systems in polygonal tundra. These degradation pathways are currently observed across the Arctic but cannot be simulated with models that do not represent spatial heterogeneities and lateral fluxes on subgrid-scales.

2. Excess ice melt, and micro- and meso-scale lateral fluxes give rise to both positive and negative feedbacks on permafrost degradation. These feedback processes and their interaction across scales control the pace of permafrost degradation in response to climate warming.

3. Projections of permafrost degradation in ice-rich lowlands are highly sensitive to the prevailing drainage conditions. Ponding of surface water entails deeper thaw and faster permafrost degradation while efficient drainage leads to shallower thaw depths and slower degradation in response to warming.

4. Multi-scale tiling models which represent topographic changes due to thermokarst activity and lateral water fluxes can explicitly simulate transient changes in the drainage conditions and hence bear the potential to more robustly project the response of ice-rich permafrost landscapes to a warming climate than simplistic one-dimensional models.

*Code availability.* The model code and settings used for the simulations are permanently deposited at https://doi.org/10.5281/zenodo. 4095341. The code is published under the GNU General Public License v3.0.

## Appendix A:  Details on the study area

Samoylov Island (72.36972 °N, 126.47500 °E) is located in the central southern part of the Lena River delta within the lowland tundra vegetation zone and features cold continuous permafrost (mean annual ground temperature of about −9 °C (Boike et al., 2019)), which has been warming rapidly in recent years (about 0.9 °C per decade (Biskaborn et al., 2019)). The island belongs to the first terrace of the Lena River delta which formed during the Holocene (Schwamborn et al., 2002). The fluvial

deposits contain substantial amounts of excess ice, mainly in form of ice wedges up to $10\,\text{m}$ deep, which can be of epigenetic and syngenetic type. The surface of the island is characterized by ice-wedge polygons and water bodies of different sizes, ranging from ponds in polygon troughs and centres to thaw lakes several hundred meters in diameter (Figure A1 a). Ice-wedge polygons show different features across different parts of the island, including mostly low-centered polygons with water-covered centres, but also water-filled troughs indicative of ice-wedge degradation, and some high-centred polygons with drained troughs. Polygons of different geomorphological type tend to occur as grouped "clusters" in the same areas of the island (see mapping by Kartoziia (2019)), and the pre-dominant types differ along gently-sloped (about $0.1\%$) transects across the island (Fig. A1 b).

## Appendix B:  Derivation of the relation between ground ice content and ground subsidence

Excess ice is defined as "the volume of ice in the ground which exceeds the total pore volume that the ground would have under natural unfrozen conditions" (NSIDC, 2020). Hence, the total volumetric ice content ($\theta_i$) of a given frozen soil layer can be separated into a pore ice content ($\theta_p$) and an excess ice content ($\theta_x$). Assuming an air fraction of zero, these volumetric ice contents and the soil matrix contents (mineral and organic) add up to one:

$$\theta_m + \theta_o + \theta_i = \theta_m + \theta_o + \theta_x + \theta_p = 1 \ . \tag{B1}$$

The excess ice scheme of CryoGrid 3 assumes, that, once a grid cell containing excess ice thaws, the entire excess ice melts and the ground volume it occupied is filled with saturated soil of the natural porosity ($\phi_{nat}$) from above-lying soil layers Westermann et al. (2016). Thus, the ground surface subsides by exactly the amount of excess ice which was contained in that cell before it thawed. Figure B1 illustrates how the content of a frozen grid cell of thickness $\Delta d$ can be partitioned into an excess ice part ($\Delta x$) and a part which reflects the soil under unfrozen conditions ($\Delta u$), i.e. with the natural porosity, such that:

$$\Delta d = \Delta s + \Delta u \ . \tag{B2}$$

The following relation for $\Delta u$ can be derived geometrically from Figure B1:

$$\Delta u = \Delta d \frac{\theta_m + \theta_o}{1 - \phi_{nat}} \tag{B3}$$
$$= \Delta d \underbrace{\frac{1 - \theta_i}{1 - \phi_{nat}}}_{\theta_p / \phi_{nat}} \tag{B4}$$

where Eq. (B1) was used to obtain the second expression. Note that $\Delta u = \Delta d$ for $\theta_i = \phi_{\text{nat}}$, i.e., if the cell contains only pore ice and no excess ice. Solving Eq. (B2) for $\Delta s$ and inserting Eq. (B4) for $\Delta u$ yields:

$$\Delta s = \Delta d - \Delta u \tag{B5}$$

$$= \Delta d - \Delta d \frac{1 - \theta_i}{1 - \phi_{\text{nat}}} \tag{B6}$$

$$= \Delta d \underbrace{\frac{\theta_i - \phi_{\text{nat}}}{1 - \phi_{\text{nat}}}}_{\theta_x} \; . \tag{B7}$$

Note that $\Delta s = 0$ if the cell contains only pore ice ($\theta_i = \phi_{\text{nat}}$), and that $\Delta s = \Delta d$ for $\theta_i = 1$, i.e. if the cell contains only excess ice and no soil matrix.

## Appendix C: Calculation of lateral surface and subsurface water fluxes

Lateral water fluxes ($q_\alpha$ [$\text{m}\,\text{s}^{-1}$]) to a tile $\alpha$ from hydrologically connected tiles $\beta \in \mathcal{N}(\alpha)$ were calculated as the sum of surface and subsurface contributions:

$$q_\alpha = q_\alpha^{\text{surf}} + q_\alpha^{\text{subs}} \tag{C1}$$

Following Darcy's law, both surface and subsurface fluxes were assumed to be proportional to a gradient in the hydraulic heads ($h$), but assuming different saturated hydraulic conductivities ($K^{\text{surf/subs}}$) and contact heights ($H^{\text{surf/subs}}$):

$$q_\alpha^{\text{surf}} = \frac{1}{A_\alpha} \sum_{\beta \in \mathcal{N}(\alpha)} K_{\alpha\beta}^{\text{surf}} \frac{h_\beta - h_\alpha}{D_{\alpha\beta}} H_{\alpha\beta}^{\text{surf}} L_{\alpha\beta} \tag{C2}$$

$$q_\alpha^{\text{subs}} = \frac{1}{A_\alpha} \sum_{\beta \in \mathcal{N}(\alpha)} K_{\alpha\beta}^{\text{subs}} \frac{h_\beta - h_\alpha}{D_{\alpha\beta}} H_{\alpha\beta}^{\text{subs}} L_{\alpha\beta} \tag{C3}$$

where $D$ is the distance between the tiles and $L$ the contact length which have been introduced in Section 2.3.3. The hydraulic head of each tile was identified with the water table elevation ($w$), or the frost table elevation ($f$) if no water table was present ($h = \max(w, f)$). The contact heights $H$ for the surface and subsurface fluxes were obtained as follows:

$$H_{\alpha\beta}^{\text{surf}} = \max(0, h^{\text{max}} - \max(s^{\text{max}}, f^{\text{max}})) \tag{C4}$$

$$H_{\alpha\beta}^{\text{subs}} = \max(0, \min(h^{\text{max}} - f^{\text{max}}, s^{\text{max}} - f^{\text{max}})) \tag{C5}$$

where $h^{\text{max}} = \max(h_\alpha, h_\beta)$ is the maximum hydraulic head, $s^{\text{max}} = \max(s_\alpha, s_\beta)$ is the maximum soil surface elevation, and $f^{\text{max}} = \max(f_\alpha, f_\beta)$ is the maximum frost table elevation of the two involved tiles. Lateral water fluxes were only applied, when both tiles were snow-free and the uppermost soil grid cell unfrozen.

## Appendix D: Definitions of the micro-topographic state in the *polygon* simulations

We largely followed the definitions of Nitzbon et al. (2019) and Nitzbon et al. (2020) to define the micro-topographic state of a three-tile polygon via the relative soil surface altitudes ($a_{\text{C/R/T}}$) and water table altitudes ($w_{\text{C/R/T}}$) of the center, rim, and trough

tiles. We distinguished between low-centred polygons (LCP), intermediate-centred polygons (ICP), high-centred polygons with inundated troughs (HCPi), high-centred polygons with drained troughs (HCPd), and water bodies (WB):

$$\text{Low-centred polygons (LCP):} \quad a_{\mathrm{C}} \leq \min\left(a_{\mathrm{R}}, a_{\mathrm{T}}\right), \tag{D1}$$

$$\text{Intermediate-centred polygons (ICP):} \quad a_{\mathrm{T}} < a_{\mathrm{C}} < a_{\mathrm{R}}, \tag{D2}$$

$$\text{High-centred polygons, inundated (HCPi):} \quad a_{\mathrm{C}} \geq \max\left(a_{\mathrm{R}}, a_{\mathrm{T}}\right) \wedge w_{\mathrm{T}} > a_{\mathrm{T}}, \tag{D3}$$

$$\text{High-centred polygons, drained (HCPd):} \quad a_{\mathrm{C}} \geq \max\left(a_{\mathrm{R}}, a_{\mathrm{T}}\right) \wedge w_{\mathrm{T}} \leq a_{\mathrm{T}}, \tag{D4}$$

$$\text{Water body (WB):} \quad \min\left(w_{\mathrm{C}}, w_{\mathrm{R}}, w_{\mathrm{T}}\right) > \max\left(a_{\mathrm{C}}, a_{\mathrm{R}}, a_{\mathrm{T}}\right). \tag{D5}$$

*Author contributions.* J.N. designed the study, conducted the numerical simulations, analyzed the results, and led the manuscript preparations. L.M and T.S.v.D. contributed to the model development. M.L., S.W. and J.B. co-designed the study. All authors interpreted the simulation results and contributed to the manuscript preparation.

*Competing interests.* The authors declare that they have no conflict of interests.

*Acknowledgements.* The authors gratefully acknowledge the Climate Geography Group at the Humboldt-Universität zu Berlin for providing resources on their high-performance-computer system. This work was supported by a grant of the Research Council of Norway (project PERMANOR, grant no. 255331). J.N. was supported by the Geo.X Research Network. M.L. was supported by a BMBF grant (project PermaRisk, grant no. 01LN1709A). S.W. acknowledges funding through Nunataryuk (EU grant agreement no. 773421).

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

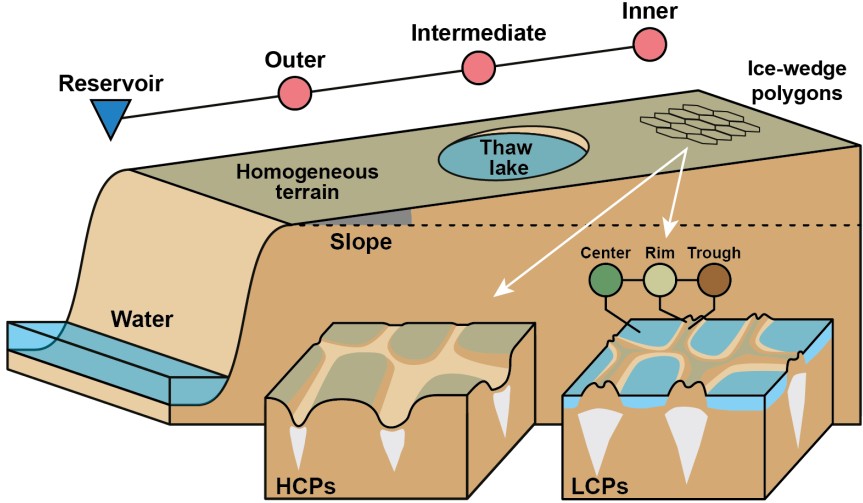

**Figure 1.** Schematic illustration of ice-rich permafrost lowlands featuring spatial heterogeneity on different scales. At the meso-scale, the terrain is gently-sloped and features larger landforms like thaw lakes. At the micro-scale, ice-wedge polygons entail a periodic patterning of the terrain which can have a low-centred (LCPs) or high-centred (HCPs) topography, depending on the grade of degradation. In this study, a tile-based modelling approach is pursued to represent these heterogeneities and investigate their effect on projections of landscape evolution and permafrost degradation.

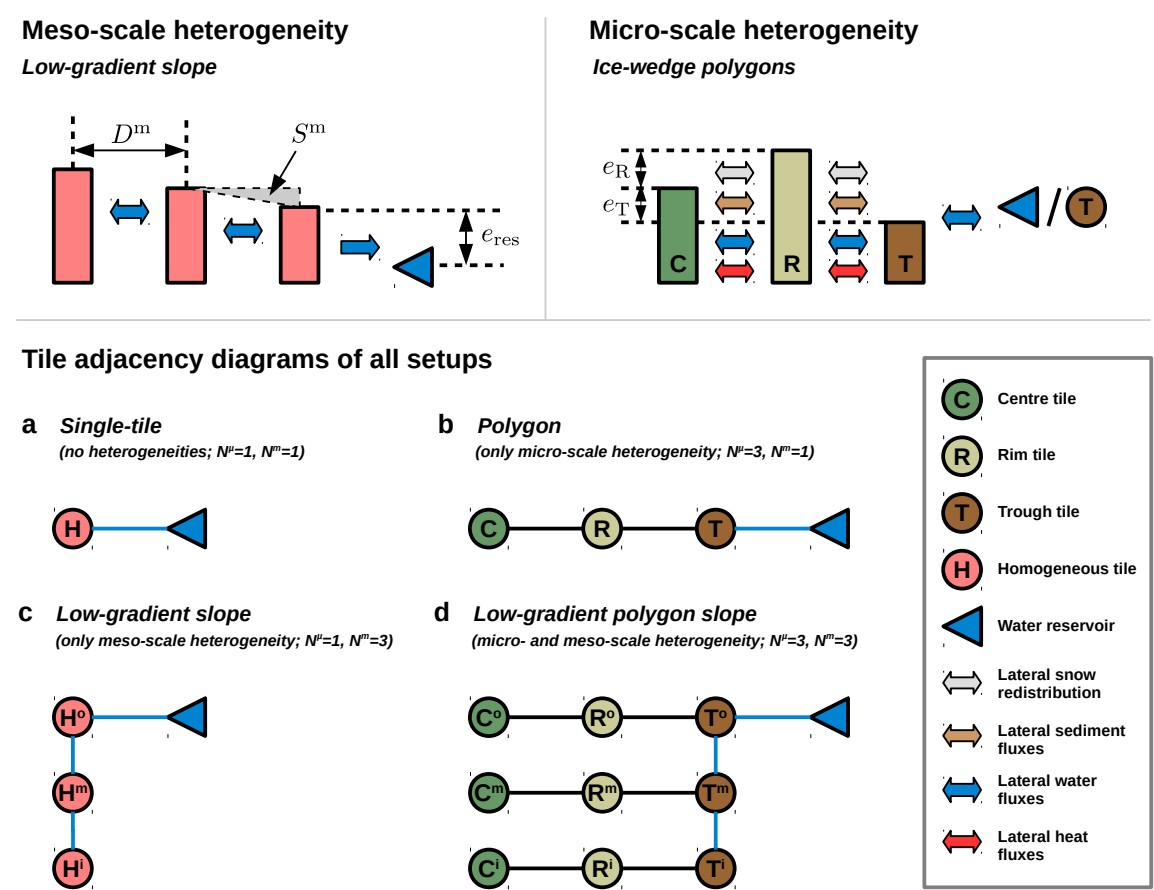

**Figure 2.** Overview of the the different tile-based model setups used to represent heterogeneity at micro- and meso-scales. An external reservoir (blue triangle) at a fixed elevation ($e_{res}$) prescribes the hydrological boundary conditions. a: The *single-tile* setup used only one tile (H) and does not reflect any subgrid-scale heterogeneity. b: The *polygon* setup reflects micro-scale heterogeneity of topography and ground ice distribution associated with ice-wedge polygons via three tiles: polygon centres (C), polygon rims (R) and inter-polygonal troughs (T). The parameters $e_R$ and $e_T$ indicate the (initial) elevation of rims and troughs relative to the center. The micro-scale tiles are interacting through lateral fluxes of heat, water, snow, and sediment. c: The *low-gradient slope* reflects meso-scale heterogeneity of an ice-rich lowland via three tiles ($H^{o,m,i}$) of which the outer one ($H^o$) is connected to a draining reservoir. The intermediate ($H^m$) and inner tiles ($H^i$) are located at distances $D^m$ along a low-gradient slope $S^m$. The meso-scale tiles are interacting through lateral surface and subsurface water fluxes. d: The *low-gradient polygon slope* incorporates heterogeneities at both micro- and meso-scales via a total of nine tiles.

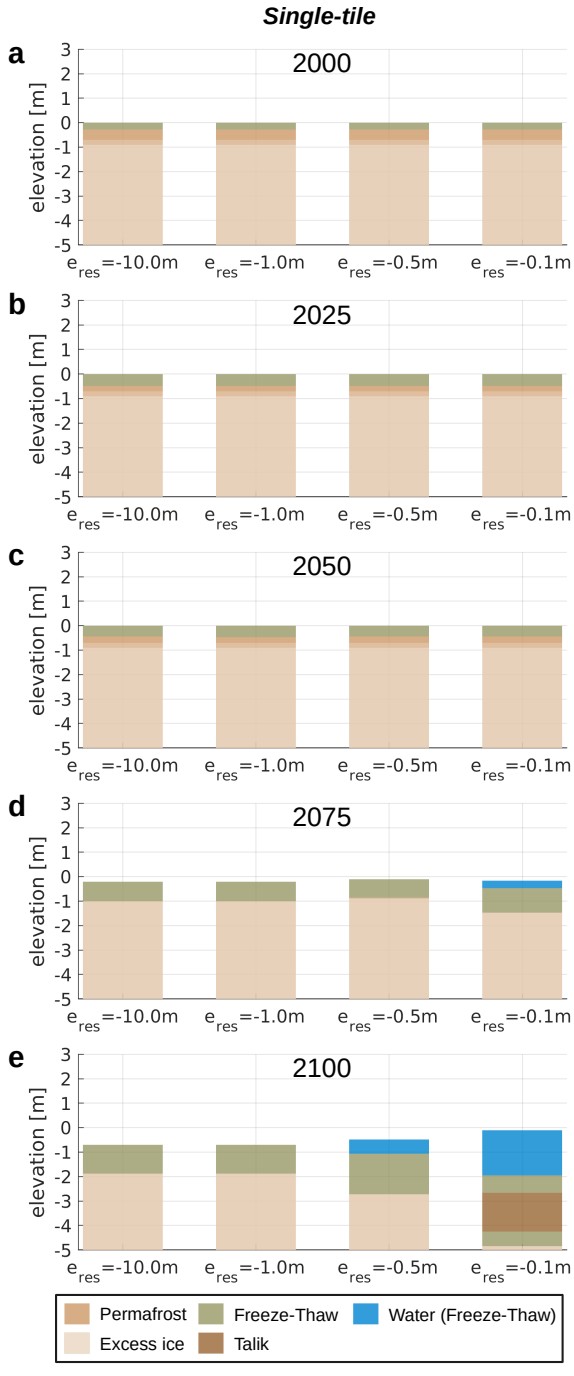

**Figure 3.** Landscape configurations in selected years (a-e) simulated with the *single-tile* setup (see Fig. 2 a) under four different hydrological boundary conditions (left to right). $e_{res}$ denotes the elevation of the external water reservoir, so that lower values correspond to better drainage.

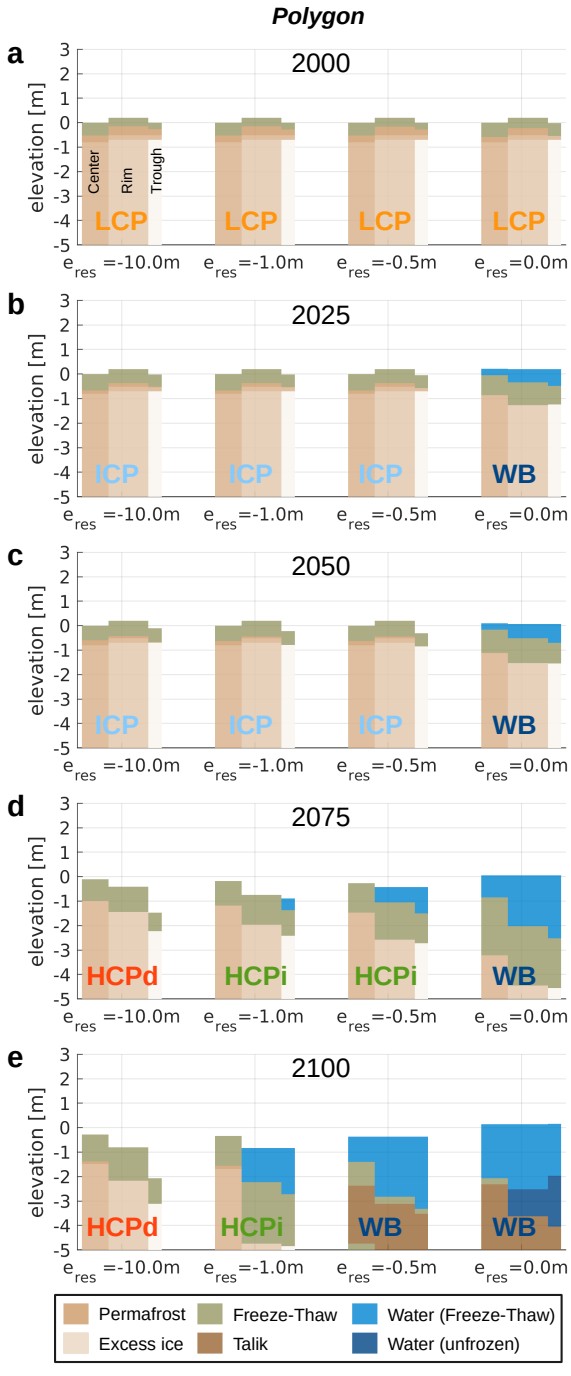

**Figure 4.** Landscape configurations in selected years (a-e) simulated with the *polygon* setup (see Fig. 2 b) under four different hydrological boundary conditions (left to right). $e_{res}$ denotes the elevation of the external water reservoir, so that lower values correspond to better drainage. The width of the three tiles corresponds to their areal fractions. Brighter colours reflect higher excess ice contents of the permafrost (see Table 3). Labels indicate the state of the micro-topography according to Eqn. (D1) to (D5).

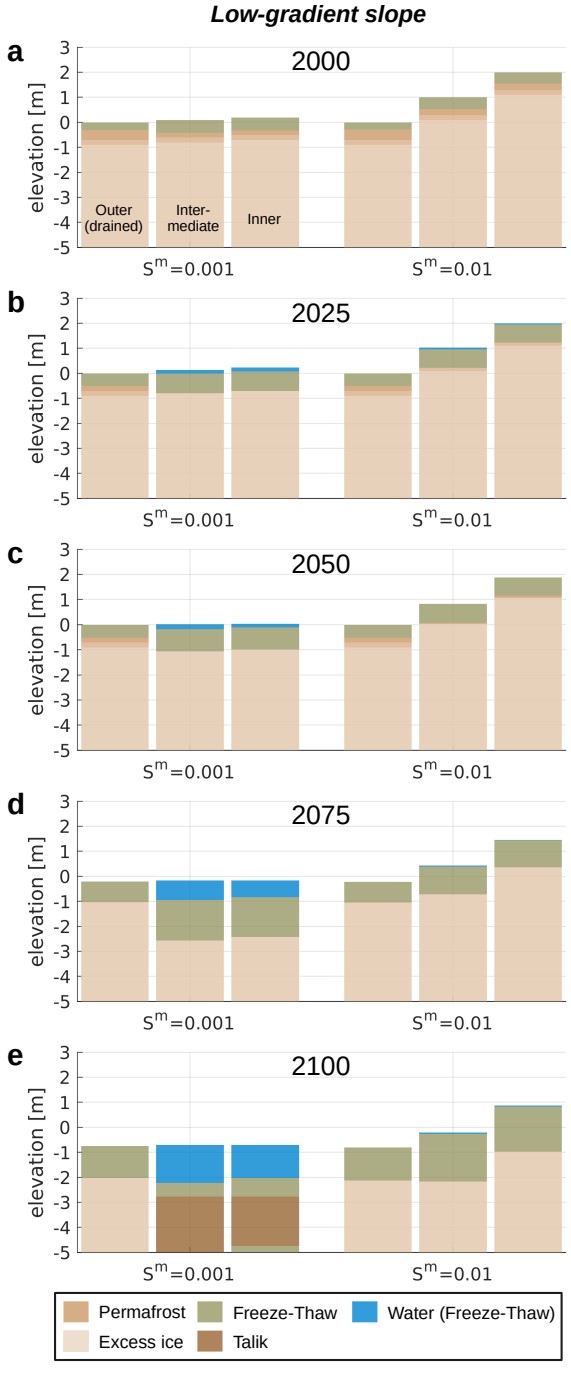

**Figure 5.** Landscape configurations in selected years (a-e) simulated with the *low-gradient slope* setup (see Fig. 2 c) for two different slope gradients ($S^m$). The "outer" tile is connected to an external reservoir ($e_{res} = -10.0$ m) which allows for efficient drainage, while the "intermediate" and "inner" tiles can drain only to adjacent tiles.

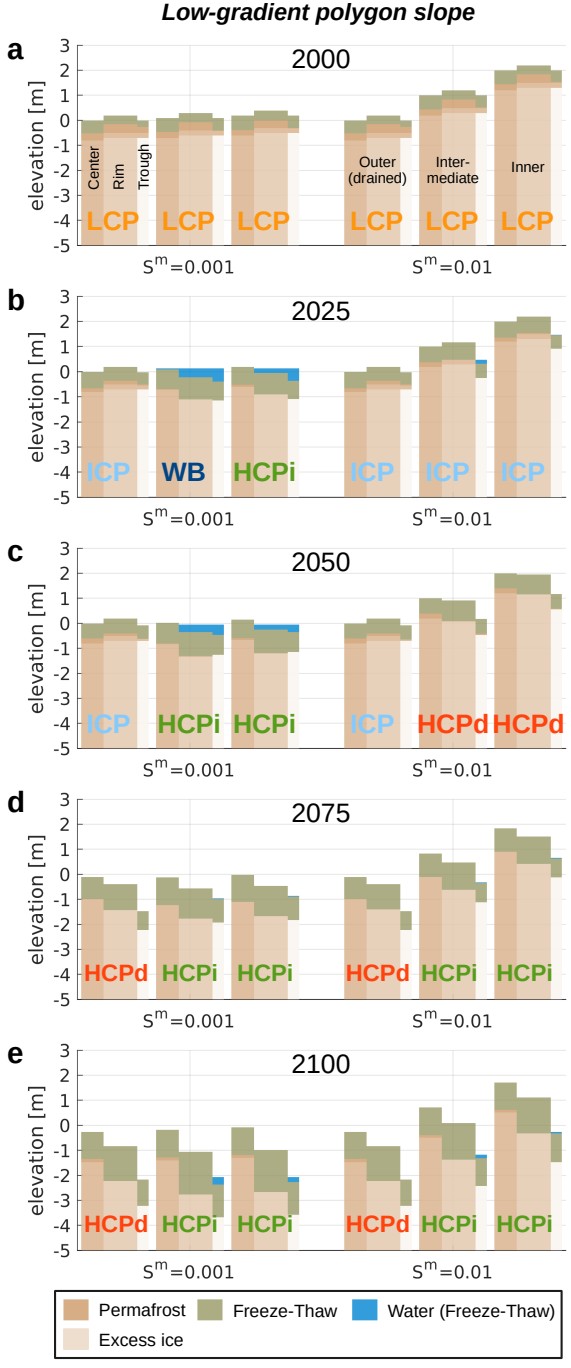

**Figure 6.** Landscape configurations in selected years (a-e) simulated with the *low-gradient polygon slope* setup (see Fig. 2 d) for two different slope gradients ($S^m$). The trough tile of the "outer" polygon is connected to an external reservoir ($e_{res} = -10.0$ m) which allows for efficient drainage. The trough tiles of adjacent polygons are connected to each other and constitute the drainage channels for the entire model domain. Brighter colours reflect higher excess ice contents of the permafrost (see Table 3). Labels indicate the state of the micro-topography according to Eqn. (D1) to (D5).

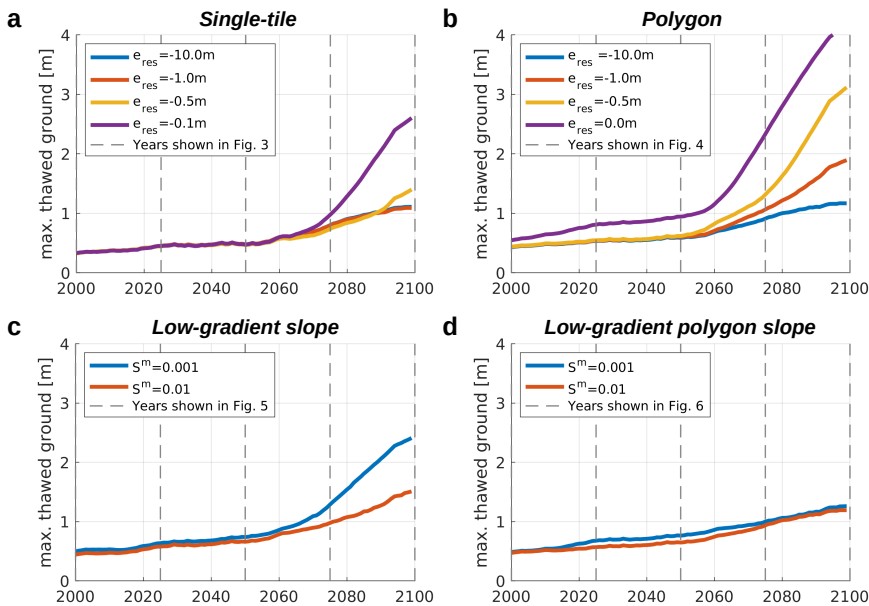

**Figure 7.** Time series of the maximum thawed ground thickness throughout the simulation period for all model setups (a to d) and parameter settings. $e_{res}$ denotes the elevation of the external water reservoir, so that lower values correspond to better drainage. $S^m$ denotes the gradient of the meso-scale slope. To derive the maximum thawed ground thickness, we first took the maximum annual thaw depth of each tile, and then averaged these, weighted according to the areal fractions of the different tiles. We then took the 11-year running mean to obtain a smoothed time series. Dashed gray lines indicate the selected years for which the landscape configuration is explicitly shown in Figures 3 to 6.

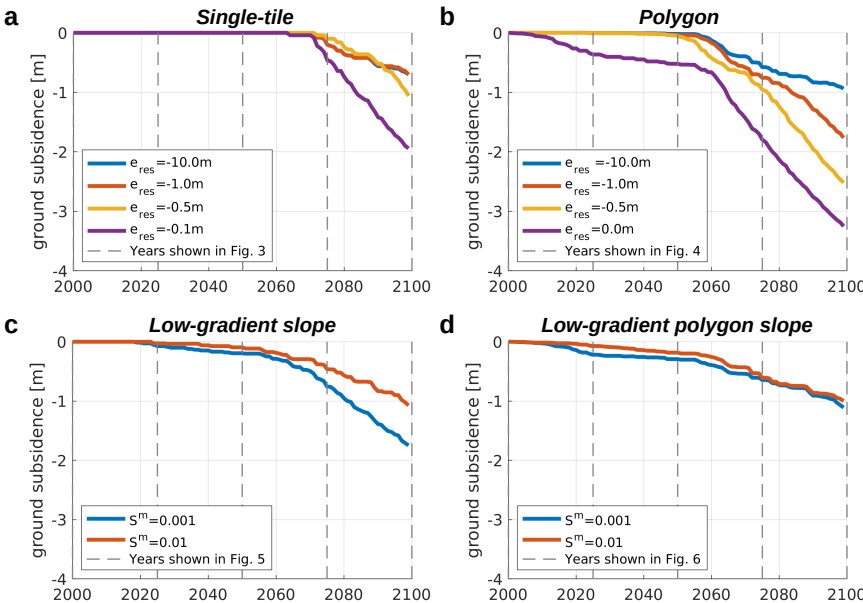

**Figure 8.** Time series of the accumulated ground subsidence throughout the simulation period for all model setups (a to d) and parameter settings. $e_{res}$ denotes the elevation of the external water reservoir, so that lower values correspond to better drainage. $S^m$ denotes the gradient of the meso-scale slope. To obtain the accumulated ground subsidence, we first took the difference between the soil surface elevation in each year and the soil surface elevation at the start of the simulation period. We then averaged these differences, weighted according to the areal fractions of the different tiles. Dashed gray lines indicate the selected years for which the landscape configuration is explicitly shown in Figures 3 to 6.

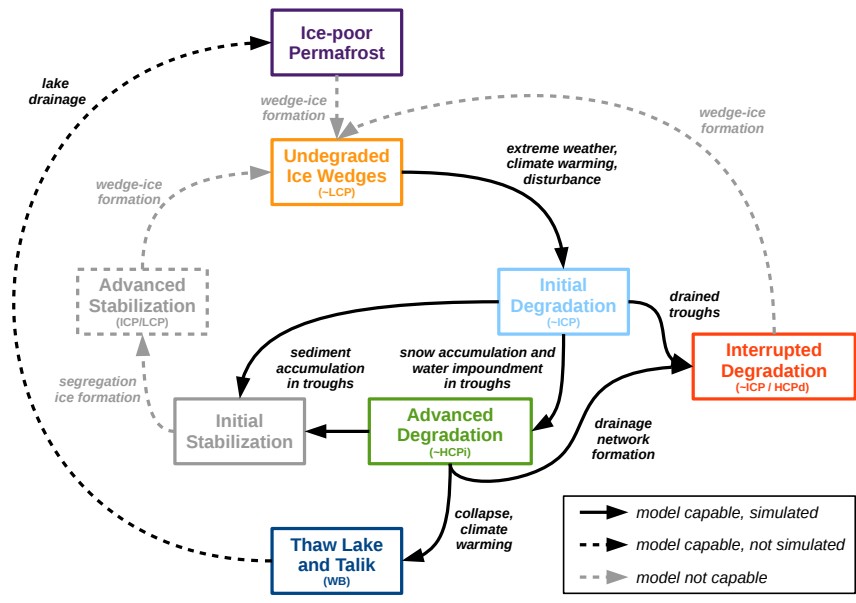

**Figure 9.** Schematic depiction of pathways of ice-rich permafrost landscape evolution as simulated within the presented model framework. The "inner" cycle reflects the cyclic evolution of ice-wedge polygons as described by Kanevskiy et al. (2017). The outer cycle involving the thaw lake stage reflects the thaw lake cylce as hypothesized by Billings and Peterson (1980). Formation of excess ice is lacking in the model such that the full cycles cannot be simulated. The expressions in brackets correspond to the simulated landscape configurations: low-centred polygons (LCP), intermediate-centred polygons (ICP), high-centred polygons with inundated troughs (HCPi), high-centred polygons with drained troughs (HCPd), and water bodies (WB). Figure adapted from Jorgenson et al. (2015) and Nitzbon et al. (2019).

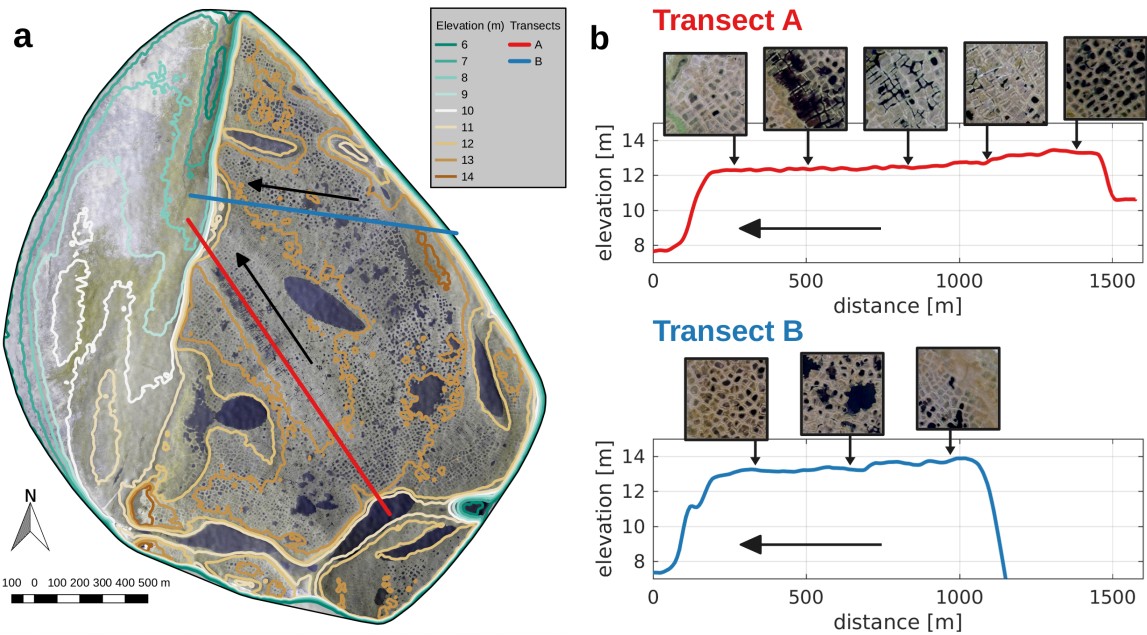

**Figure A1.** a: Aerial photography of Samoylov Island with elevation contour lines based on the ArcticDEM (Porter et al., 2018). The study area is covered with water bodies of different sizes and ice-wedge polygons of different geomorphological stages. b: Two (arbitrary) examples of transects across the island, reflecting the low-gradient sloped terrain with steep cliffs at the margins. The inlets are details (about 100 m in diameter) of the aerial photography, reflecting polygon clusters of similar type along the transects. Horizontal arrows point in the main drainage direction. Note that ice-wedge polygons show little signs of degradation in the highest-elevated parts and close to the margins of the island (first and last inlet of both Transects). Along Transect A, the abundance of thermokarst troughs increases in the downstream direction of the slope. Small water bodies are visible in the central part of Transect B.

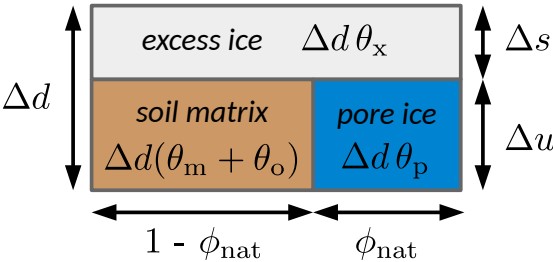

**Figure B1.** Schematic illustration of an ice-rich soil layer of thickness $\Delta d$ which is composed of excess ice ($\theta_x$), pore ice ($\theta_p$), and a soil matrix ($\theta_m + \theta_o$).

**Table 1.** Overview of the terminology used in this manuscript to refer to the spatial scale of permafrost landscape features and processes.

| Terminology | Superscript | Length scale | Examples |
|---|---|---|---|
| subgrid | – | $\lesssim 10^5\,\mathrm{m}$ | all mentioned below |
| micro | $\mu$ | $\simeq 10^0\text{–}10^1\,\mathrm{m}$ | Ice-wedge polygons, Hummocks |
| meso | m | $\simeq 10^2\text{–}10^3\,\mathrm{m}$ | Thermo-erosion catchments, Thaw lakes |
| macro | – | $\simeq 10^4\text{–}10^5\,\mathrm{m}$ | River delta, LSM grid cell |

**Table 2.** Generic soil stratigraphy used to represent the subsurface composition of all tiles. An ice-rich layer of variable ice content ($\theta_i$) is located at variable depth ($d_x$) from the surface. Note that the effective excess ice content ($\theta_x$) is linked to the total ice content ($\theta_i$) and the natural porosity ($\phi_{nat}$) via the relation given in Equation (2). The soil texture is used to parametrize the freezing-characteristic curve of the respective layer (see Westermann et al. (2016) for details).

| Depth from [m] | Depth to [m] | Mineral $\theta_m$ | Organic $\theta_o$ | Nat. por. $\phi_{nat}$ | Soil texture | Water/ice $\theta_w^0$ | Comment |
|---|---|---|---|---|---|---|---|
| 0 | 0.1 | 0 | 0.15 | 0.85 | sand | 0.85 | Vegetation layer |
| 0.1 | 0.2 | 0.10 | 0.15 | 0.75 | sand | 0.75 | Organic layer |
| 0.2 | $d_x - 0.2$ | 0.25 | 0.10 | 0.65 | silt | 0.65 | Mineral layer |
| $d_x - 0.2$ | $d_x$ | 0.20 | 0.15 | 0.55 | sand | 0.65 | Intermediate layer |
| $d_x$ | 10 | $\frac{1.05 - \theta_i}{2}$ | $\frac{0.95 - \theta_i}{2}$ | 0.55 | sand | $\theta_i$ | Variable excess ice layer |
| 10 | 30 | 0.50 | 0.05 | 0.45 | sand | 0.45 | Ice-poor layer (Taberit) |
| 30 | 1000 | 0.90 | 0 | 0.10 | sand | 0.10 | Bedrock |

**Table 3.** Overview of the model parameters for different representations of the micro-scale. Note that on average the *polygon* tiles (C,R,T) feature the same excess ice content and depth as the *homogeneous* tile (H). Setting the area of the homogeneous tile to $A^{\mu} = 1\,\mathrm{m}^2$ is an arbitrary choice, as it does not affect the magnitude of the lateral fluxes due to the assumption of translational symmetry.

| Parameter | Symbol | Unit | C | R | T | area-weighted mean | sum | H |
|-----------|--------|------|---|---|---|--------------------|-----|---|
| Stratigraphy | | | | | | | | |
| Depth of excess ice layer | $d_x$ | [m] | 1.0 | 0.9 | 0.7 | 0.9 | – | 0.9 |
| Ice content of excess ice layer | $\theta_i$ | [-] | 0.65 | 0.75 | 0.95 | 0.75 | – | 0.75 |
| Topography | | | | | | | | |
| Areal fraction | $\gamma$ | [-] | $\frac{1}{3}$ | $\frac{1}{2}$ | $\frac{1}{6}$ | – | 1 | 1 |
| Total area | $A^{\mu}$ | [m²] | 46.7 | 70 | 23.3 | – | 140 | 1 |
| Initial elevation | $e$ | [m] | 0.0 | 0.2 | 0.0 | 0.1 | – | 0.0 |

**Table 4.** Overview of the parameter variations in the simulations which were conducted for the four different model setups (see Figure 2). Note that the last value for $e_{\text{res}}$ in the *single-tile* and *polygon* setups – which reflects poorly-drained conditions – was chosen to be 0.1 m less than the mean initial elevation of the respective setting (see Table 3).

| Setup name | Number of tiles ($N$) | Parameter varied | Parameter values | Figures |
|---|---|---|---|---|
| *Single-tile* | 1 | $e_{\text{res}}$ | $-10.0$ m; $-1.0$ m; $-0.5$ m; $-0.1$ m | 3, 7a, 8a |
| *Polygon* | 3 | $e_{\text{res}}$ | $-10.0$ m; $-1.0$ m; $-0.5$ m; $0.0$ m | 4, 7b, 8b |
| *Low-gradient slope* | 3 | $S^{\text{m}}$ | $0.001$; $0.01$ | 5, 7c, 8c |
| *Low-gradient polygon slope* | 9 | $S^{\text{m}}$ | $0.001$; $0.01$ | 6, 7d, 8d |