# Peer review of "Effects of multi-scale heterogeneity on the simulated evolution of ice-rich permafrost lowlands under a warming climate"

_The Cryosphere, 2020_

## Referee Comment (RC1) · Anonymous Referee #1 · 27 Jun 2020

Artic permafrost is considered one of the key tipping elements of the Earth system. However, researchers face the problem that modelling studies and observations show that the dynamics in permafrost affected regions often depend on abrupt, non-linear processes that are locally very confined, while quantifying the resulting impacts on the global climate requires using low resolution models which do not account for these small scale processes.

Here, Nitzbon et al. propose a tiling approach that allows representing surface hetero-geneities – namely the polygonal structures typical for many permafrost regions and low gradient slopes with a length scale of ∼100m – in the CryoGrid model. They use

the model to investigate the effects of 21st-century warming (RCP8.5) and demonstrate that their approach is capable of capturing subgrid-scale variations in the resulting degradation of permafrost. Thus, the proposed approach could potentially facilitate the understanding of high-latitude processes and improve their representation in Earth System models.

In general, the study presents highly relevant work in an important field and, overall, the manuscript is well written. Especially the introduction-, discussion, -conclusion and outlook sections help the reader place the study in the context of previous and future research on permafrost-affected regions. However, while I think that the proposed tiling approach could present an important step in improving coarse-resolution models as well as our understanding of high latitude landscapes, the authors do not demonstrate this in their work. As it stands, the manuscript only shows that the approach adds to the model's complexity, but fails to provide compelling evidence that this results in an actual improvement of the simulations. Here, the paper requires major revisions before it can be considered for publication.

General comments

1) As stated above, my main concern with the manuscript is that the authors do not compare any aspect of their simulations to observations or to simulations with any other point-scale model that has been validated in the past. Here, the authors claim that they investigate a site on Samoylov and even state that there is a large amount of observational data available for the island that can be used to validate numerical models. However, they make no use of this data making it impossible for the reader to judge whether the tiling approach leads to results that are closer to reality.

2) It may be difficult to evaluate the model's performance even with the data that is available on Samoylov. However, in this case the results need to be described in a manner that allows the reader to understand how the newly implemented processes change the model's behaviour. In this way, the reader has at least the chance to judge

whether the behaviour of the model is plausible. In the results section, the authors merely present the landscape evolution for different setups without providing any details on the underlying mechanisms or explanations as to what causes the differences in the simulations. This is not only true for the more complex cases that involve subgrid-scale heterogeneities and later exchanges between the tiles, but even for the very basic one-tile setups. For example, while reading, I was always wondering why the permafrost degradation was so much faster in the poorly-drained than in the well-drained setup? Is it because of higher heat conductivity of water? Or is it an albedo effect due to wetter soils and due to the formation of surface water bodies? Admittedly, some details are provided in the discussion section, but this is nowhere near enough to understand what the model actually does.

3) If it is partly the aim of the paper to present the approach to large-scale modellers as a way of improving their parametrizations, it requires a better verbal description of the scheme's benefits. To exaggerate a bit: One could look at figure 3 and 4 and decide that actually the simple homogenous, well-drained setup does surprisingly well when compared to the complex polygon-landscape setup. In 2100, I find subsidence of roughly 1m, an active layer depth of about 1m and largely unchanged ground below, which is very close to what I get when I aggregate the three tiles of the complex setup. Admittedly, the simple scheme misses the water bodies (especially between 2025 – 2050), but they also seem to be quite small. The same is true when I compare 5a and b as well as 5c and d. There is very little in the paper that convinces the reader that the (overall) landscape evolution can't be simulated well with a single-tile setup with an appropriate set of soil parameters. I do believe that the scheme presents an important improvement, but that point needs to be made more clearly in the manuscript.

Specific comments

P.4, L.87 ff: Study area – To my understanding the study is more a demonstration of technical possibilities of your developments rather than an investigation that relies on the specific setup of Samoylov or on data from the island. Therefore I would suggest

to leave out the entire description (subsection 2.2 ) of the study area, because it is a bit misleading in two ways:

A) Expectations – With such a detailed description of the site, one expects to find a comparison to observations from Samoylov at a later part of the manuscript. B) Technical capabilities of the tiling approach – The tiling approach is not really capable of representing specific (complex) heterogeneous landscapes. With no real information of the actual spatial distribution of the tiles within the encompassing grid box, any tiling approach only ever represents a well mixed setting which is not really the case for the island.

I think it is sufficient to state that the initial soil conditions, forcing data and areal fractions were chosen based on Samoylov and that stratigraphy represents a generic profile based on previous studies of the island, without giving more information on the site. However, the ideal way would be to use any of the available observations from Samoylov to validate your model – then the information you provide would be very welcome.

P.6, L.117 ff: You provide the units for density and specific heat but not for heat capacity and conductivity. Other parameters are also introduced without unit later in the text. In my opinion it wouldn't be a problem to leave out the units altogether, however if you are nice enough to provide them, this should be done consistently.

P.6, L.123: What does the "effectively" refer to?

P.6, L.125 ff: How does the model deal with the surface water? Is there a (water) depth dependant runoff-formulation and does the water evaporate? Or does it simply pool until it can infiltrate?

P.6, L.127: Is the field capacity the same for the organic and mineral soil?

P.6, L.145: Why this formulation for the effective h. conductivity?

P.7, L.164: is > are

P.7, L.171: I think the information that you are not treating meso-scale lateral heat fluxes should be a bit more prominent – e.g. in the abstract you say that your model captures lateral heat fluxes at the scales not captured by ESMs which implies that also the meso-scale heat fluxes are represented.

P.8, Figure 2: A very nice figure that gives a very intuitive overview over your setups. Maybe you could separate the vertical subplots more distinctly from the connection/network diagrams to make it clearer that these are two different aspects and that the vertical setups in a and b are also applicable in subplot c and d. Also the information with respect to dx and the ice content is slightly confusing because it is only shown in subfigure a – I think it could be left out from the plot.

P.9, L.181: What happens to the vegetation layer in the case that surfaces are inundated for longer periods?

P.9, Table2: Has the column "Water" been mentioned before?

P.10 Table3: The legend states that the average of the polygonal setup is equal to the single tile setup, however this does not seem to be the case for the Reservoir elevation (poorly drained).

P.11, L.219: What is a "repeatedly appended base climatological period" ?

P.11, L.224: Why only the two extreme cases for the single-tile setup? I think it could also be helpful to see the behaviour for a medium-drainage constellation?

P.12, L.244 ff: Why is the degradation rate so much faster in the poorly than in the well-drained setting? Heatconductivity/Capacity, albedo? A description of the underlying mechanisms would be very helpful.

P.12, L.250 ff: Could you explain the cause of the diverging behaviour in the tiles, it appears to be similar to the differences between the well- and poorly drained single tile setups. Also why is the outer tile behaving very differently while the inner and intermediate tiles behave similarly? The figure seems to indicate that the same dynamics

could be obtained with a two-tile setup (inner tile / outer tile).

P.16, L.302 ff. | P.17, L.310 ff: It seems as if around the year 2060-2080 the rate of active-layer deepening (rate of ground subsidence) increases in all setups, which you also note on page 17 l. 312 . What is the reason for this non-linear behaviour? Is this related to the forcing? If so could you maybe provide a timeseries of the forcing – e.g. 2m temperatures?

P.16, L.303: Which sub-grid scale interactions result in the active layer being deeper in the landscape simulation – at least until the year 2090? Based on the soil properties one would expect it to be a combination of the well- and poorly drained single tile setup?

P.18, L.358 ff: Maybe these explanations – at least between lines 358 and 364 – fit better in the results section.

P.18, L.371 ff: What does "become more involved" mean in this context?

P.19, L.382 ff: Again, this information may be better suited for the result section. As an afterthought on sections 4.1 and 4.2: maybe it is possible to disentangle the process description and the interpretation to how this relates to other studies? Because the descriptive parts would fit extremely well into the results section?

P.19 L.401: Here, one could argue that your results actually show the opposite: When including both meso and micro-scale processes, the results are actually fairly close to the single tile (well drained) set-up. Thus, setting the soil parameters adequately may already be enough for projections with large-scale models.

P21, Figure 6: A very nice overview figure. It would be nice though if you could explain the abbreviations in the caption.

P. 21, .L447: from > form

P.22, L.453: I wouldn't say that this is specific to northeast Siberia but rather a simple

test case.

P.22, L.462 ff: I am not convinced that you demonstrated that in this study: There is no comparison to observations or a detailed description of the subgrid-scale processes. Thus you present a new (and I do believe more suitable) approach, but you do not show how this helps reduce uncertainties.

P22, L.472 ff: Here, I do not agree with the authors' conclusion. On one hand they merely show that their approach increases complexity but not that this complexity improves the quality of the results and provides further constraints on projections of future permafrost degradation. On the other hand they do not show that their approach is suitable for the ESMs, i.e. that the approach can be scaled to the respective resolutions. With respect to permafrost-affected regions, one important issue would be that the dependencies in the model are sufficiently linear, allowing subgrid-scale heterogeneity to be represented by one (or a few) parameter set(s) that represent an average over large areas. Personally, I do believe that the scheme presents an important improvement, but that point needs to be made much more clearly in the manuscript.

P. 22, L.477 ff: I think it would greatly increase the quality of the study if the authors could provide some validation or evaluation of the model. There is neither a detailed description of the processes that lead to the results, nor have any aspects of the simulation been compared to observations or to simulations with any other point-scale model.

---

## Referee Comment (RC2) · Anonymous Referee #2 · 24 Aug 2020

Authors implement micro- and meso-scale spatial grid resolution in the 1D CryoGrid model to illustrate the spatial effect of microtopographic feature on the rate of permafrost thaw. Authors found that implementing higher spatial resolution in the model leads to "more realistic possibilities (L13)". Now sure what type of possibilities they have in mind? Improving spatial representation of the polygonal tundra in the ESM type models is important. However, the current version of the manuscript lacks clarity. I found it hard to follow and the central Figure 2 looks like an electrical circuit diagram. If the take-home message is that every ESM needs to have micro-, meso-scale permafrost tundra representation, then it needs to be clearly stated. Maybe including recommendations on how authors think that can be done easily, in their opinion, using

the current approach. Overall, this is a timely and important work that needs to be published. However, the description, terminology, and flow require more work. I have a hard time reading and understanding the concept laid in the paper. I understand that much of the tiling concept was introduced in previous work (Aas et al.). However, the recap could be extremely helpful in setting up the stage in this study. Also, talking about uncertainties between different tiling approaches might be useful too. For example, if we average the overall effect from individual polygons, it could have the same carbon footprint as representing the polygonal tundra heterogeneity in one tile. When could that be or not be true? The comments below illustrate the lack of my knowledge of the presented scaling method. I hope the authors would not be discouraged by my comments and try to help me better understand their work in the revised version of the manuscript.

Abstract Can be shortened and cleaned. There are too many we found..., also found..., for example..., our results suggest... It was really hard to wrap my head around what exactly was found and how that helps science, stakeholders, economy, etc.

L50. What is tile-based modeling approach? Need to define.

L77. "To quantify the sensitivity". I am not sure how the sensitivity was addressed?

L79. What type of sources of uncertainty? See my main comments. By making super refined models, we can introduce many small uncertainties which will superposition at the end. The question is, where is the golden ratio?

Table 1. should it be m2? Are we talking about the gridcell resolution?

Figure 1. Are you simulating the transect or an entire area? If you model an entire area, then how that area is going to look under different resolutions? In an ideal case, we should be able to take any area and then apply a deferent resolution to it (zooming in and out). The different surface features will be more/less pronounced based on

spatial resolution. Then we can model future changes under different resolutions and the difference between modeling results should tell us how fine we should go. This way sounds more straight forward to me...

L127 what is field capacity?

If I understand it correctly, the $\theta$_i is initialized? I suggest to rename $\Delta$p to $\Delta$d_ice. Typically, p represent pressure. So, ice thickness is initialized too? Does the model start from the initialized ice thickness or there is a steady-state run? So, the second term in equation 2 should be less than or equal to 1? Otherwise subsidence could be greater than the ice thickness. Can that be the case? I did not understand the denominator. What is 1-phi_{nat} mean?

L149. Need a reference after "... hierarchical approach".

N̂$\mu$ is that somewhat standard notation? I had a hard time following that notation and remembering what it means. Is there a way to change it or use some other more intuitive notation? For example, use 1m2 or 1km2 notations. What is the total area modeled? Is this modeling represent a transect or a 2d area?

Does homogeneous means that one tile represents the entire transect. If so, then it would be easier to say that 1 tile approach. What is the external reservoir? Is that water table depth?

Figure 2, I had a hard time to understand and follow.

L204 How many topological characteristics were used? Are these characteristics represent only magnitude of the lateral fluxes or something else too?

Table 3 what is 'not a null' over 'big sigma' columns represent? I like the results section and was able to make more sense of it. I think that discussing the geomorphological processes as well as figure 6 diversify the message: "the importance of the tile-approach adoption by the ESM type models." I guess, it is important to focus on that message instead of diving into the concepts and pathways of the polygonal tundra

geomorphological evolution.

For this type of paper, I would like to see a more in-depth mathematical analysis of the difference between different spatial resolutions as well as discussion of the corresponding uncertainties. I understand that this might lead to way too much work and may not be feasible in this paper. Then I suggest to exclude the ESM modeling discussion from the article and give it a different angle from the beginning. Consider bringing Figure 6 into the methods or introduction. Then it will setup the stage for the follow-up story.

---

## Author Comment (AC1) · 29 Sep 2020

We thank the reviewer for the detailed review of our manuscript and the constructive feedback on our work. We provide answers to the comments below.

Reviewer comments are in *blue italics*.

Author replies are in normal font.

Extracts from the manuscript are in **bold**, and modifications in the revised manuscript are **highlighted yellow**.

*Anonymous Referee #1*

*Artic permafrost is considered one of the key tipping elements of the Earth system. However, researchers face the problem that modelling studies and observations show that the dynamics in permafrost affected regions often depend on abrupt, non-linear processes that are locally very confined, while quantifying the resulting impacts on the global climate requires using low resolution models which do not account for these small scale processes.*

*Here, Nitzbon et al. propose a tiling approach that allows representing surface heterogeneities – namely the polygonal structures typical for many permafrost regions and low gradient slopes with a length scale of ~100m – in the CryoGrid model. They use the model to investigate the effects of 21st-century warming (RCP8.5) and demonstrate that their approach is capable of capturing subrgid-scale variations in the resulting degradation of permafrost. Thus, the proposed approach could potentially facilitate the understanding of high-latitude processes and improve their representation in Earth System models.*

*In general, the study presents highly relevant work in an important field and, overall, the manuscript is well written. Especially the introduction-, discussion, -conclusion and outlook sections help the reader place the study in the context of previous and future research on permafrost-affected regions.*

*However, while I think that the proposed tiling approach could present an important step in improving coarse-resolution models as well as our understanding of high latitude landscapes, the authors do not demonstrate this in their work. As it stands, the manuscript only shows that the approach adds to the model's complexity, but fails to provide compelling evidence that this results in an actual improvement of the simulations. Here, the paper requires major revisions before it can be considered for publication.*

We appreciate that the reviewer acknowledges the relevance of our study and identifies the potential significance of our work with respect to an improved understanding of permafrost landscape dynamics as well as the improvement of coarse-resolution models. We understand that the major point of criticism of the reviewer is that we do not demonstrate that our model

developments result in an actual improvement of projections of 21$^{st}$-century permafrost degradation. We can understand the concerns of the reviewer, but we are confident that our model developments and numerical simulations constitute a substantial advancement compared to previous works. With the explicit representation of micro- and meso-scale landscape heterogeneity, our model facilitates simulation of permafrost landscape dynamics and feedbacks on permafrost degradation in an unprecedented way. We think that a clarification and reformulation of the scope and objectives of our study as well as an extended discussion of the model's advantages over more simple models will rule out this major criticism.

We further agree with the reviewer in that our simulations demonstrate the potential of the multi-scale tiling approach to improve coarse-resolution models. It was, however, not our main objective to prove this in the present study. Instead, our major goal was to show that our approach is capable of representing a wide range of different landscape evolution and permafrost degradation pathways which are known to occur in ice-rich permafrost lowlands. In particular, we investigated which effect the incorporation of micro- and meso-scale heterogeneities has on the simulated landscape evolution and permafrost degradation. Importantly, we do not claim that the most complex model configuration necessarily provides the most accurate projections. However, we do want to convey the insight that potentially important feedback mechanisms can only be represented by those model configurations which take into account micro- and/or meso-scale heterogeneities.

In the revised version of the article, we reworked the introduction section to point out that we do not primarily aim at improving coarse-resolution model projections, but that our study should be considered an explorative modelling exercise of potential feedbacks due to landscape heterogeneity on different spatial scales. We revised the text in different places, for example our objectives are now stated more clearly:

**The overall scope of this study is to investigate the effect of micro- and meso-scale heterogeneities on the transient evolution of ice-rich permafrost lowlands under a warming climate. Specifically, we addressed the following objectives:**

**1. To identify degradation pathways and feedback processes associated with lateral fluxes on the micro- and meso-scale.**

**2. To quantify permafrost degradation in terms of thaw-depth increase and ground subsidence in dependence of the representation of micro- and meso-scale heterogeneities.**

We further clarified these objectives by the following explanations:

**Overall, our goal is to provide a scalable framework for exploring the evolution of permafrost landscapes in response to a warming climate, which could potentially be incorporated into LSMs to allow more robust projections of permafrost loss in response to climate change. The presented simulations should thus be considered as numerical experiments to identify important scales and controls of permafrost degradation.**

With these modifications we hope to have clarified the scope of our study. In addition, we revised the Discussion section of our article such that it more clearly states the potentials and advantages of our approach compared to more simple models.

The detailed comments of the reviewer are addressed in a point-by-point fashion below.

**General comments**

*1) As stated above, my main concern with the manuscript is that the authors do not compare any aspect of their simulations to observations or to simulations with any other point-scale model that has been validated in the past. Here, the authors claim that they investigate a site on Samoylov and even state that there is a large amount of observational data available for the island that can be used to validate numerical models. However, they make no use of this data making it impossible for the reader to judge whether the tiling approach leads to results that are closer to reality.*

We agree that we do not present any comparison of our simulations with observational data from the field site in the present manuscript. However, in a preceding study (Nitzbon et al. 2019), it has been shown that the micro-scale tiling approach which was also used in this study, is capable of reflecting the heterogeneity of thermal, hydrological, and snow characteristics associated with polygonal tundra micro-topography. That study involved a comprehensive evaluation of the model using field observations.

The main reason for not presenting further model evaluation in the present study is that it was not the aim of the study to prove that the presented tiling framework necessarily enables a more accurate reproduction of measurements. Instead, our study primarily aims at identifying qualitative effects of subgrid-heterogeneity on the projections of landscape evolution and permafrost degradation. A second reason is that the available data (soil temperatures, thaw depths, etc.) from the field site on Samoylov Island only capture the micro-scale heterogeneity associated with ice-wedge polygons, while no long-term data are available which document the meso-scale variability of these parameters (Boike et al. 2019).

In the revised version of the manuscript, we clarified that it is not the goal of our study to provide quantitatively accurate simulations, but rather to explore which pathways of landscape evolution can be retraced by different model configurations (see reply above). We would further like to stress that the simulated landscape evolution in the model qualitatively corresponds well with established knowledge on thermokarst landscape dynamics and observations from across the Arctic (e.g. Kokelj et al. 2013, Liljedahl et al. 2016). A qualitative evaluation of our modelling is provided in Sections 4.1 and 4.3 of the revised manuscript.

*2) It may be difficult to evaluate the model's performance even with the data that is available on Samoylov. However, in this case the results need to be described in a manner that allows the reader to understand how the newly implemented processes change the model's behaviour. In this way, the reader has at least the chance to judge whether the behaviour of the model is plausible. In the results section, the authors merely present the landscape evolution for*

*different setups without providing any details on the underlying mechanisms or explanations as to what causes the differences in the simulations. This is not only true for the more complex cases that involve subgrid-scale heterogeneities and later exchanges between the tiles, but even for the very basic one-tile setups. For example, while reading, I was always wondering why the permafrost degradation was so much faster in the poorly-drained than in the well-drained setup? Is it because of higher heat conductivity of water? Or is it an albedo effect due to wetter soils and due to the formation of surface water bodies? Admittedly, some details are provided in the discussion section, but this is nowhere near enough to understand what the model actually does.*

We thank the reviewer for acknowledging the difficulties to evaluate the performance of the presented model framework with the scarce data available at Arctic field sites. We appreciate the suggestion to overcome this shortcoming by putting more effort into explaining the model dynamics such that it becomes possible for the reader to retrace the model behaviour and to judge its plausibility. We would like to note, however, that some of these explanations have been provided already in previous studies using the CryoGrid 3 model (Westermann et al. 2016, Langer et al. 2016, Nitzbon et al. 2019, Nitzbon et al. 2020), and that we wanted to avoid presenting these as novel findings.

For the revised version of the manuscript, we thoroughly extended the results section by explanations of the model dynamics and feedbacks and provided explanations which were previously contained in the Discussion section. We also restructured the Results section which is now sorted according to the different model setups, which increase in complexity. By this, it is easier for the reader to discern the feedback processes which are relevant to the different model configurations (i.e., the representation of heterogeneities).

*3) If it is partly the aim of the paper to present the approach to large-scale modellers as a way of improving their parametrizations, it requires a better verbal description of the scheme's benefits. To exaggerate a bit: One could look at figure 3 and 4 and decide that actually the simple homogenous, well-drained setup does surprisingly well when compared to the complex polygon-landscape setup. In 2100, I find subsidence of roughly 1m, an active layer depth of about 1m and largely unchanged ground below, which is very close to what I get when I aggregate the three tiles of the complex setup. Admittedly, the simple scheme misses the water bodies (especially between 2025 – 2050), but they also seem to be quite small. The same is true when I compare 5a and b as well as 5c and d. There is very little in the paper that convinces the reader that the (overall) landscape evolution can't be simulated well with a single-tile setup with an appropriate set of soil parameters. I do believe that the scheme presents an important improvement, but that point needs to be made more clearly in the manuscript.*

As we have stated above, our goal was to demonstrate that the model is capable of capturing pathways and feedbacks of permafrost landscape evolution which are not possible to reflect in simple single-tile setup. While we agree with the reviewer, that it might be possible to simulate a similar total amount of permafrost degradation (by 2100) in the most complex setup to that in an appropriate single-tile setup, the multi-tile setup shows a different transient

landscape evolution and captures subgrid-scale processes which are potentially relevant in other realms like biogeochemistry. For example, we stress the potential relevance of subgrid-scale heterogeneities for carbon decomposition, to which also small-scale landscape features can contribute substantially (e.g., Abnizova et al. 2012, Langer et al. 2015).

In the revised version of the manuscript, we discuss the benefits of the tiling method more extensively in the discussion Section 4.2 and put our modeling work in the context of approaches from the LSM/ESM community. Overall, our study is intended to inform coarse-scale modelers about the effects of including heterogeneities at different scales, and thus to enable them to decide whether and how these could be implemented in their model frameworks.

*Specific comments*

*P.4, L.87 ff: Study area – To my understanding the study is more a demonstration of technical possibilities of your developments rather than an investigation that relies on the specific setup of Samoylov or on data from the island. Therefore I would suggest to leave out the entire description (subsection 2.2 ) of the study area, because it is a bit misleading in two ways:*

*A) Expectations – With such a detailed description of the site, one expects to find a comparison to observations from Samoylov at a later part of the manuscript.*

*B) Technical capabilities of the tiling approach – The tiling approach is not really capable of representing specific (complex) heterogeneous landscapes. With no real information of the actual spatial distribution of the tiles within the encompassing grid box, any tiling approach only ever represents a well mixed setting which is not really the case for the island.*

*I think it is sufficient to state that the initial soil conditions, forcing data and areal fractions were chosen based on Samoylov and that stratigraphy represents a generic profile based on previous studies of the island, without giving more information on the site. However, the ideal way would be to use any of the available observations from Samoylov to validate your model – then the information you provide would be very welcome.*

We agree with the reviewer in this point and hence removed the extensive description of the study area from the main text. Instead, we provide a shortened version of the study area description as appendix A. We decided to keep the former Figure 1 in the appendix (now Figure A1) as we think that it gives a good impression of the abundance and scales of heterogeneities and landforms common to real-world permafrost lowlands. In the main text, we added an schematic figure (new Figure 1) which illustrates ice-rich permafrost lowlands with the micro- and meso-scale heterogeneities addressed in this study. We think that this new Figure 1 underlines the conceptual character of our modelling study instead of raising expectations which cannot be met.

*P.6, L.117 ff: You provide the units for density and specific heat but not for heat capacity and conductivity. Other parameters are also introduced without unit later in the text. In my opinion it wouldn't be a problem to leave out the units altogether, however if you are nice enough to provide them, this should be done consistently.*

We consistently provide the units of all physical quantities in the revised version of the manuscript.

*P.6, L.123: What does the "effectively" refer to?*

The change of the snow density can only change due to infiltration and refreezing, but not due to internal processes in the snowpack. We deleted the word „effectively" as it is admittedly confusing and did not contain additional information.

*P.6, L.125 ff: How does the model deal with the surface water? Is there a (water) depth dependant runoff-formulation and does the water evaporate? Or does it simply pool until it can infiltrate?*

The hydrology scheme allows for evaporation of surface water and it can run off laterally, either to an adjacent tile, or into an „external reservoir", if the tile is connected to one (see Figure 2). We clarified this point in the revised manuscript:

**Excess water is allowed to pond above the surface, leading to the formation of a surface water body. ==Surface water is modulated by evaporation as well as lateral fluxes to adjacent tiles or into an external reservoir (see *Lateral fluxes* below).==**

*P.6, L.127: Is the field capacity the same for the organic and mineral soil?*

Yes, the field capacity parameter is identical for all soil layers. A sensitivity study in a previous study revealed that the overall model dynamics are not sensitive to this parameter (see Supplementary Information to Nitzbon et al. (2020)).

*P.6, L.145: Why this formulation for the effective h. conductivity?*

The lateral fluxes into the external reservoir are calculated based on a Darcy approach as described in Nitzbon et al. (2019). The „effective" hydraulic conductivity incorporates the distance ($D$) and contact length ($L$) between the respective tile and the reservoir. For a tile which can drain in all directions, the effective hydraulic conductivity is obtained as follows: $K_{eff}=K*P/D=K*2\pi*D/D=2\pi K$. This is further explained in the SI of Nitzbon et al. (2020).

*P.7, L.164: is > are*

Thanks. Corrected in the revised version.

*P.7, L.171: I think the information that you are not treating meso-scale lateral heat fluxes should be a bit more prominent – e.g. in the abstract you say that your model captures lateral heat fluxes at the scales not captured by ESMs which implies that also the meso-scale heat fluxes are represented.*

We agree with this concern and added a sentence in the model description to justify this simplification:

We did not consider lateral fluxes of heat, snow and sediment at the meso-scale, as these were assumed to be negligible on the time scale of interest (heat, sediment), or too uncertain (snow).

We also revised the formulation in the abstract. We mention the potential effect of meso-scale lateral heat fluxes in the Discussion (Section 4.1):

Previous modeling studies have also demonstrated that the stability and the thermal regime of permafrost in the vicinity of thaw lakes is affected by meso-scale lateral heat fluxes from taliks forming underneath the lakes (Rowland et al., 2011; Langer et al., 2016). These effects have not been considered in this study.

*P.8, Figure 2: A very nice figure that gives a very intuitive overview over your setups. Maybe you could separate the vertical subplots more distinctly from the connection/network diagrams to make it clearer that these are two different aspects and that the vertical setups in a and b are also applicable in subplot c and d. Also the information with respect to dx and the ice content is slightly confusing because it is only shown in subfigure a – I think it could be left out from the plot.*

We thank the reviewer for appreciating the added value of this figure. It is correct that the vertical cross-sections for the setups a and b are also applicable to the setups c and d, respectively. We revised this figure according to the suggestions of the reviewer. To clarify the different setups, we also revised the names assigned to the different setups in a way which we hope is more intuitive to understand.

*P.9, L.181: What happens to the vegetation layer in the case that surfaces are inundated for longer periods?*

The organic-rich vegetation layer does not change when the surface is inundated for longer periods. While the vegetation type would probably adapt to the aquatic conditions in reality, we assume that the thermal properties of this layer would not change substantially.

*P.9, Table2: Has the column "Water" been mentioned before?*

The column refers to the initial water/ice content. The water content in the unfrozen part of the ground is, however, modified by the hydrology scheme. We adopted the label of the column to clarify this.

*P.10 Table3: The legend states that the average of the polygonal setup is equal to the single tile setup, however this does not seem to be the case for the Reservoir elevation (poorly drained).*

The statement in the legend was indeed confusing. True is, that the depth of the excess ice layer and the excess ice content of the homogeneous tile correspond to the area-weighted mean of the three polygon tiles. In addition, the reservoir elevation of the poorly-drained setup was set 0.1m below the mean initial elevation of the surface. To clarify this point, we revised Table 3 and added a new Table 4 which gives an overview of the parameter variations.

*P.11, L.219: What is a "repeatedly appended base climatological period" ?*

The anomalies from the CCSM4 projections for the period after 2014 were applied to a fifteen-year „climatological base period" (2000-2014). The formulation has been clarified in the revised manuscript.

*P.11, L.224: Why only the two extreme cases for the single-tile setup? I think it could also be helpful to see the behaviour for a medium-drainage constellation?*

The idea behind considering only the two extreme cases for the hydrological boundary conditions was to create a direct link to the preceding study by Nitzbon et al. (2020), in which these boundary conditions were treated as confining extreme cases. We agree that, as a stand-alone independent work, the present study provides more insights, if simulations under intermediate hydrological conditions were considered as well. Thus, we conducted simulations for two additional intermediate levels of the external reservoir ($e_{res}$=-0.5m and

$e_{res}$=-1.0m) for both the single-tile and the polygon setups. The additional simulation results are described and discussed in the revised manuscript, in which the Results section is now structured more clearly according to the different model setups.

*P.12, L.244 ff: Why is the degradation rate so much faster in the poorly than in the well-drained setting? Heat conductivity/Capacity, albedo? A description of the underlying mechanisms would be very helpful.*

The degradation rate is primarily controlled by the thaw depth which is in turn affected by the hydrological regime of the active layer. On the one hand, thawed saturated soil has a higher thermal conductivity than drained soil, which allows higher ground heat fluxes and hence deeper thaw. On the other hand, ice-rich soil layers need more heat to thaw than ice-poor soil layers due to the higher latent heat content. These and other counteracting effects establish a non-trivial relationship between the hydrological regime of the active layer, and the annual (maximum) thaw depth (e.g., Atchley et al., 2016). Simulations with CryoGrid 3 typically show that wetter conditions cause deeper thaw depths and hence faster degradation (e.g., Nitzbon et al., 2019, Martin et al., 2019). This is particularly the case when surface water bodies form, since these alter the surface energy balance (e.g., lower albedo) and have a high heat capacity, which delays the refreezing and can favour the development of taliks.

In the revised manuscript, the dependency of degradation rate on the drainage conditions is further elaborated on through presentation of further simulations for intermediate drainage conditions and more extensive explanations in the main text. For example, we added the following paragraphs in section 3.1:

==Overall, the simulation results indicate that permafrost degradation is strongest as soon as a limitation of water drainage results in the formation of a surface water body. The presence of surface water changes the energy transfer at the surface in different ways. First, it reduces the surface albedo, resulting in a higher portion of incoming shortwave radiation. Second, water bodies have a high heat capacity which slows down their freeze-back compared to soil. As a last point to mention, the thawed saturated deposits beneath the surface water body have a higher thermal conductivity compared to unsaturated deposits, which allows heat to be transported more efficiently from the surface into deeper soil layers. These findings are consistent with previous CryoGrid 3 simulations for ice-wedge polygons (Nitzbon et al., 2019) and peat plateaus (Martin et al., 2019).==

==During the initial phase of excess ice melt which occurs between 2050 and 2075, our simulations suggest a non-monotonous dependence of permafrost degradation on the drainage conditions. [...] This can likely be attributed to contrasting effects of the hydrological regime on thaw depths. When the near-surface ground is unsaturated [...], the highly-porous organic-rich surface layers have an insulating effect on the ground below due their low thermal conductivity. On the other hand, less heat is required to melt the ice contained in the mineral soil layers whose ice content corresponds to the field capacity, than if their pore space was saturated with ice. In the intermediate case==

**with $e_{res}$=-0.5m, the combination of dry, insulating near-surface layers and ice-saturated mineral layers beneath leads to the lowest thaw depths and hence the slowest initial permafrost degradation. However, as soon as a surface water body forms in that simulation (between 2075 and 2100), the positive feedback on thaw described above takes over, resulting in stronger degradation by 2100 compared to the well-drained settings ($e_{res}$=-1.0m and $e_{res}$=-10.0m) for which no surface water body forms during the simulation period.**

*P12, L.250 ff: Could you explain the cause of the diverging behaviour in the tiles, it appears to be similar to the differences between the well- and poorly drained single tile setups. Also why is the outer tile behaving very differently while the inner and intermediate tiles behave similarly? The figure seems to indicate that the same dynamics could be obtained with a two-tile setup (inner tile / outer tile).*

The diverging behaviour of the outer tile from the inner ones can be explained by the fact that it is connected to a low-lying reservoir and hence well-drained, while the inner two tiles can only be drained via the outer tile. As the slope has a very low gradient (0.1%), this drainage is not very efficient and causes water to impound in the inner tiles, which in turn accelerates degradation due to the feedbacks discussed for the single-tile setup. In the revised version of the manuscript, we added the following explanations in section 3.3:

*Irrespective of the slope gradient, the simulated evolution of the outer tiles is very similar to that of the well-drained single-tile simulations throughout the entire simulation period [...] The similarity to the well-drained single-tile simulations can be explained by the fact that the outer tile is very efficiently drained, such that the lateral water input from the intermediate tile is directly routed further into the external reservoir. Hence, the "upstream" influence on the outer tile becomes negligible.*

It is furthermore correct, that a two-tile setup would likely result in a similar pattern. However, we decided to use a three-tile setup for several reasons:

- It was not clear a priori that the inner tiles would develop so similar, since the intermediate tile is closer to the drainage point of the slope. At the same time it is affected by water input from the inner tile which lies upstream. Hence it was of interest to us to see which of these effects would dominate.

- The variability of ice-wedge polygon types along transects on Samoylov Island (Figure A1 in the revised manuscript) suggested that degradation along the slope could be stronger than in the innermost part lying upstream.

- For consisency, we wanted to use the same number of tiles to represent heterogeneities on the micro- and meso-scale.

In fact, there are slight but significant differences between the "intermediate" and "inner" tiles which are explained in the revised manuscript (Section 3.3).

To present a broader picture of possible degradation pathways along low-gradient slopes, we conducted additional simulations for the two slope setups (homogeneous and polygon), where we set the slope gradient to 1.0%. These simulations reveal further insights which are presented, explained, and discussed in the revised version of the manuscript (Sections 3.3 and 3.4)

*P.16, L.302 ff. | P.17, L.310 ff: It seems as if around the year 2060-2080 the rate of active-layer deepening (rate of ground subsidence) increases in all setups, which you also note on page 17 l. 312 . What is the reason for this non-linear behaviour? Is this related to the forcing? If so could you maybe provide a timeseries of the forcing – e.g. 2m temperatures?*

The nonlinear increase in the degradation rate after 2060 or so is an important observation. The effect is noticeable irrespective of the specific model setup and has similarly been reported in previous studies using the same forcing data (Westermann et al., 2016, Langer et al., 2016, Nitzbon et al., 2020). We think that this effect can be explained by multiple factors: first, the meteorological forcing results in exceptionally high thaw depths during the 2060s, which initiates positive feedbacks to the excess ice melt (snow accumulation, water impoundment). Second, around that time the soil has warmed to a level, where residual liquid water critically slows down the back-freezing. In combination these effects cause a nonlinear shift in the degradation rate for most settings around the 2060s. While this effect can likely be generalized, the timing is strongly related to the forcing and hence the location of the study area. In the revised manuscript, we mention and discuss this effect in section 3.2:

**[...] We explain the acceleration of permafrost degradation at the beginning of the second half of the simulation period (Figures 7 b and 8 b, purple lines) by a combination of additional warming from the meteorological forcing, and positive feedbacks due to the surface water body (as explained for the single-tile simulation in Section 3.1).**

We decided to not include a time series of the forcing data, as we attribute the effect mainly to a non-linearity in the ground thermal dynamics. The temperature forcing does not show a non-linear increase during that period.

*P.16, L.303: Which sub-grid scale interactions result in the active layer being deeper in the landscape simulation – at least until the year 2090? Based on the soil properties one would expect it to be a combination of the well- and poorly drained single tile setup?*

This is an important observation which deserves further explanation. In the poorly-drained single-tile simulations, water can drain from the system as soon as the water table is above

-0.1m relative to the surface. Therefore, the drainage is still more efficient than for the two inner tiles of the three-tile slope setup, where drainage is very inefficient. For these tiles, surface water formation occurs much earlier than in the single-tile simulations and hence higher thaw depths are observed earlier (see answer and modifications mentioned above). In other words, the „poorly-drained" setting does not really reflect the most extreme case which would be a water-logged setting where runoff is precluded. This, however, would result in physically unrealistic situations where surface water would accumulate over multiple years, since evaporation is consistently lower than precipitation.

*P.18, L.358 ff: Maybe these explanations – at least between lines 358 and 364 – fit better in the results section.*

We agree and moved these explanations to the results section where we now provide further detailed explanations of the model dynamics.

*P.18, L.371 ff: What does "become more involved" mean in this context?*

We wanted to express that the dynamics become more complicated. We changed the wording in the revised manuscript.

*P.19, L.382 ff: Again, this information may be better suited for the result section. As an afterthought on sections 4.1 and 4.2: maybe it is possible to disentangle the process description and the interpretation to how this relates to other studies? Because the descriptive parts would fit extremely well into the results section?*

We agree with the reviewer and followed this suggestion in the revised manuscript.

*P.19 L.401: Here, one could argue that your results actually show the opposite: When including both meso and micro-scale processes, the results are actually fairly close to the single tile (well drained) set-up. Thus, setting the soil parameters adequately may already be enough for projections with large-scale models.*

Here, we wanted to express that the inclusion of micro- and/or meso-scale heterogeneities can result in permafrost degradation rates which are not reflected by single-tile simulations. For example, the polygon simulations consistently showed an earlier onset and stronger rate of permafrost degradation than the respective single-tile simulations. We agree, however, that this statement was imprecise and could be misinterpreted, such that we revised our formulation.

*P21, Figure 6: A very nice overview figure. It would be nice though if you could explain the abbreviations in the caption.*

The caption has been extended accordingly for the revised version. In addition, the distinction between high-centred polygons with inundated and drained troughs has been refined and is also indicated in Figures 5-8.

*P.21, .L447: from > form*

Thanks. Typo has been corrected.

*P.22, L.453: I wouldn't say that this is specific to northeast Siberia but rather a simple test case.*

We agree and changed the formulation accordingly.

*P.22, L.462 ff: I am not convinced that you demonstrated that in this study: There is no comparison to observations or a detailed description of the subgrid-scale processes. Thus you present a new (and I do believe more suitable) approach, but you do not show how this helps reduce uncertainties.*

We agree that claiming a reduction of uncertainty is not necessarily supported by the data we show in the study. However, we have shown that our approach allows us to simulate degradation pathways which correspond to observations from across the Arctic. Thus, they bear the potential for more realistic site-level assessments. We modified our conclusions for the revised manuscript.

*P22, L.472 ff: Here, I do not agree with the authors' conclusion. On one hand they merely show that their approach increases complexity but not that this complexity improves the quality of the results and provides further constraints on projections of future permafrost degradation. On the other hand they do not show that their approach is suitable for the ESMs, i.e. that the approach can be scaled to the respective resolutions. With respect to permafrost-affected regions, one important issue would be that the dependencies in the model are sufficiently linear, allowing subgrid-scale heterogeneity to be represented by one (or a few) parameter set(s) that represent an average over large areas. Personally, I do believe that the scheme presents an important improvement, but that point needs to be made much more clearly in the manuscript.*

As stated in previous points, we see the major contribution of our work in the possibility to simulate pathways and feedbacks of permafrost landscape evolution in an unprecedentedly realistic way. The tiling method allows to do this without the need to increase the grid resolution and hence computational costs drastically. Here, we do not suggest that our approach could be adopted 1:1 in coarse-scale LSM/ESM frameworks, but only state that our works contributes to the development of such model frameworks. For example, Aas et al. 2019 have demonstrated the applicability of the coupled tiles to represent polygonal tundra and peat plateaus in the Noah-MP LSM.

We carefully revised our conclusions in the revised manuscript, thereby also taking into account the additional simulations that we conducted. With respect to the implications of our study for coarse-scale modellers, we extended the Discussion in Section 4.2 and rephrased the criticized conclusion.

*P.22, L.477 ff: I think it would greatly increase the quality of the study if the authors could provide some validation or evaluation of the model. There is neither a detailed description of the processes that lead to the results, nor have any aspects of the simulation been compared to observations or to simulations with any other point-scale model.*

We agree with the reviewer that the validation and evaluation of the model is an important issue. Site-level studies which apply the presented model framework to different sites and compare the simulations to observational data, are thus highly desirable as future steps. However, suitable long-term datasets, particularly of ground subsidence, constitute a strong limitation to such endeavours. In addition, we would like to point out that the CryoGrid 3 model has already been applied in different contexts and other study areas, and those studies put a stronger focus on quantitative evaluations (Martin et al., 2019, Nitzbon et al., 2019, Schneider von Deimling et al., 2020, Zweigel et al., 2020). The need for suitable field data for model evaluation has been stressed in the revised Outlook section.

References (not contained in the original manuscript)

Abnizova, A., Siemens, J., Langer, M., & Boike, J. (2012). Small ponds with major impact: The relevance of ponds and lakes in permafrost landscapes to carbon dioxide emissions. Global Biogeochemical Cycles, 26(2). https://doi.org/10.1029/2011GB004237

Atchley, A. L., Coon, E. T., Painter, S. L., Harp, D. R., & Wilson, C. J. (2016). Influences and interactions of inundation, peat, and snow on active layer thickness: Influence of Environmental Conditions on ALT. Geophysical Research Letters, 43(10), 5116–5123. https://doi.org/10.1002/2016GL068550

Langer, M., Westermann, S., Walter Anthony, K., Wischnewski, K., & Boike, J. (2015). Frozen ponds: Production and storage of methane during the Arctic winter in a lowland tundra landscape in northern Siberia, Lena River delta. Biogeosciences, 12(4), 977–990. https://doi.org/10.5194/bg-12-977-2015

Schneider von Deimling, T., Lee, H., Ingeman-Nielsen, T., Westermann, S., Romanovsky, V., Lamoureux, S., Walker, D. A., Chadburn, S., Cai, L., Trochim, E., Nitzbon, J., Jacobi, S., & Langer, M. (2020). Consequences of permafrost degradation for Arctic infrastructure – bridging the model gap between regional and engineering scales. The Cryosphere Discussions, 1–31. https://doi.org/10.5194/tc-2020-192

---

## Author Comment (AC2) · 29 Sep 2020

We thank the reviewer for evaluating our manuscript, and for the numerous comments which help to improve it. We provide answers to the comments below.

Reviewer comments are in *blue italics*.

Author replies are in normal font.

Extracts from the manuscript are in **bold**, and modifications in the revised manuscript are **highlighted yellow**.

**Anonymous Referee #2**

**Received and published: 24 August 2020**

Authors implement micro- and meso-scale spatial grid resolution in the 1D CryoGrid model to illustrate the spatial effect of microtopographic feature on the rate of permafrost thaw. Authors found that implementing higher spatial resolution in the model leads to "more realistic possibilities (L13)". Now sure what type of possibilities they have in mind? Improving spatial representation of the polygonal tundra in the ESM type models is important. However, the current version of the manuscript lacks clarity. I found it hard to follow and the central Figure 2 looks like an electrical circuit diagram. If the take-home message is that very ESM needs to have micro-, meso-scale permafrost tundra representation, then it needs to be clearly stated. Maybe including recommendations on how authors think that can be done easily, in their opinion, using the current approach.

Overall, this is a timely and important work that needs to be published. However, the description, terminology, and flow require more work. I have a hard time reading and understanding the concept laid in the paper. I understand that much of the tiling concept was introduced in previous work (Aas et al.). However, the recap could be extremely helpful in setting up the stage in this study. Also, talking about uncertainties between different tiling approaches might be useful too. For example, if we average the overall effect from individual polygons, it could have the same carbon footprint as representing the polygonal tundra heterogeneity in one tile. When could that be or not be true? The comments below illustrate the lack of my knowledge of the presented scaling method. I hope the authors would not be discouraged by my comments and try to help me better understand their work in the revised version of the manuscript.

We appreciate that the reviewer acknowledges the importance and timeliness of our work. From the comments we understand that the main criticism of the reviewer is, that the description of our methodological approach lacks clarity and that the implications of our work with respect to large-scale models are not sufficiently explained. We think that these issues could be related to the fact that we presupposed that readers would be familiar with preceding works in which the concept of laterally coupled tiles was applied in a permafrost context (Langer et al., 2016, Aas et al., 2019, Nitzbon et al., 2019, and Nitzbon et al., 2020). In these papers, the concept is introduced and applied to investigate different subgrid-scale processes in permafrost ecosystems. In particular, Nitzbon et al. (2019) introduced a three-tile setup for ice-wedge polygonal tundra and evaluated it using field observations from Samoylov Island in the Lena River delta. In the present study, we use that setup to represent micro-scale heterogeneity of ice-rich lowlands, and extended it by a representation of meso-scale heterogeneity. We show that the combined representation of micro- and meso-scale heterogeneities gives rise to pathways and feedbacks of landscape evolution which qualitatively agree with observations, but which have not been simulated with a numerical model before.

In order to underline the original and novel contributions of the present study, we carefully revised the manuscript and took care that it stands for its own. In particular, we revised the objectives, extended the description of the tiling approach, reworked the figures, conducted additional simulations, restructured and extended the results sections, and extended the discussion with respect to large-scale modelling. Hereafter, we address the specific comments of the reviewer in a point-by-point style.

Abstract Can be shortened and cleaned. There are too many we found. . ., also found. . ., for example. . ., our results suggest. . . It was really hard to wrap my head around what exactly was found and how that helps science, stakeholders, economy, etc.

We agree that the abstract contained a lot of detailed information. We revised and shortened it for the revised manuscript.

**L50. What is tile-based modeling approach? Need to define.**

We added a paragraph in the Introduction which introduces the "tiling approach" and summarizes previous work:

Many of the above-referenced modelling studies employed a so-called "tiling approach" to account for subgrid-scale heterogeneities of permafrost terrain. Instead of discretizing extensive landscape domains on a high-resolution mesh, the landscape is partitioned into a low number of characteristic landscape units, each of which is associated with a representative "tile" in the model. Thereby, geometrical characteristics (e.g., areas, distances, perimeters) are used to parameterize lateral fluxes among the landscape units. For example, Langer et al. (2016) used a two-tile model setup to investigate the effect of lateral heat fluxes in a lake-rich permafrost landscape, and Nitzbon et al. (2019) suggested a three-tile model setup to represent the micro-scale heterogeneity associated with ice-wedge polygon tundra. Schneider von Deimling et al. (2020) applied a five-tile setup to represent the interaction of linear infrastructure such as roads with underlying and surrounding permafrost. To date, the tiling method has not been applied to simultaneously represent permafrost landscape heterogeneities and their interactions across multiple spatial scales. Moreover, we extended the description of the tiling scheme in the Methods section so that the methodology is now understandable without being familiar with the tiling concept in advance:

We used the concept of laterally coupled tiles to represent subgrid-scale spatial heterogeneities of permafrost terrain (Langer et al., 2016; Aas et al., 2019; Nitzbon et al., 2019). In general, the tiling concept involves the partitioning of real-world landscapes into a certain number of characteristic units, which are associated with the major surface and subsurface heterogeneities found in the landscape. Each of these units is then represented by a single ``tile'' in a permafrost model, and multiple tiles can interact through lateral exchange processes. The tiling approach thus allows to simulate subgrid-scale heterogeneities and lateral fluxes in macro-scale models like LSMs/ESMs, without discretizing extensive landscape domains on a high-resolution mesh, thereby keeping computational costs at a reasonable level.

In addition to these modifications in the main text, we think that the new Figure 1 as well as the revised Figure 2 facilitate a more intuitive understanding of our modelling approach.

**L77. "To quantify the sensitivity". I am not sure how the sensitivity was addressed?**

We understand that this formulation is imprecise. We wanted to express that our objective was to investigate how the projected permafrost degradation is affected by different representations of subgrid-heterogeneities. We rephrased the objectives which now read as follows:

**Specifically, we address the following objectives:**

1. To investigate the transient evolution of ice-rich permafrost landscapes in response to climate warming using different representations of micro- and meso-scale heterogeneity, thereby identifying degradation pathways and feedback processes associated with lateral fluxes on these different spatial scales.

2. To quantify the sensitivity of projected permafrost thaw and ground subsidence to different representations of micro- and meso-scale heterogeneity.

*L79.* What type of sources of uncertainty? See my main comments. By making super refined models, we can introduce many small uncertainties which will superposition at the end. The question is, where is the golden ratio?

We understand that this formulation is imprecise. In a preceding study, Nitzbon et al. (2020) found that different hydrological boundary conditions can lead to large deviations in the projections of ice-rich permafrost thaw under the RCP8.5 warming scenario. Here, we wanted to investigate whether these deviations are reduced, when heterogeneities on the meso-scale

are represented in the model. We removed this sentence from the manuscript as might be misleading, and modified the respective paragraph which now reads:

Overall, our goal is to provide a scalable framework for exploring the evolution of permafrost landscapes in response to a warming climate, which could potentially be incorporated into LSMs to allow more robust projections of permafrost loss in response to climate change. The presented simulations should thus be considered as numerical experiments to identify important scales and controls of permafrost degradation, instead of providing accurate site-specific projections.

**Table 1. should it be m2? Are we talking about the gridcell resolution?**

The table is only intended to introduce the terminology with respect to spatial scales. The length scales are given in the unit [m] and the numbers reflect the order of magnitude of landscape features on the respective length scale. The numbers do not correspond to a grid cell resolution of the employed model. We hope that this becomes more clear with the extended description of the tiling approach.

Figure 1. Are you simulating the transect or an entire area? If you model an entire area, then how that area is going to look under different resolutions? In an ideal case, we should be able to take any area and then apply a deferent resolution to it (zooming in and out). The different surface features will be more/less pronounced based on spatial resolution. Then we can model future changes under different resolutions and the difference between modeling results should tell us how fine we should go. This way sounds more straight forward to me...

The figure is primarily intended to give an impression of the various types and scales of landscape heterogeneities in permafrost lowlands. The transects are intended to give an impression of the variability of the ice-wedge polygons along low-gradient slopes. In light of the critique of reviewer #1 regarding this figure (and the entire section on the study area), we understand that the figure might raise wrong expectations regarding the scope of our modelling work. In fact we do not intend to simulate the landscape evolution for Samoylov Island, but rather consider generic test cases in which we vary the representation of micro-and meso-scale heterogeneities. For this, we use the tile-based modelling approach which involves some assumptions on the geometry of the modelled landscape. To clarify the scope of our study, we replaced Figure 1 with a schematic illustration of a generic ice-rich lowland landscape. In this new figure, we also indicated how the different tiles represent different parts of the overall landscape. The revised Figure 1 (in addition with the revised Figure 2), should make the scope of the study and the modelling approach more intuitive to understand. We moved the Figure of Samoylov Island to the Appendix (Figure A1), as it provides a real-world example for the simulated permafrost lowlands.

**L127 what is field capacity?**

Field capacity refers to the capacity of the soil to "hold" water after drainage of excess water. In CryoGrid3, there is a parameter which specifies the volumetric water content which the soil takes upon infiltration. We modified the respective formulation:

Infiltrating water is instantaneously routed downwards through unfrozen soil layers, whose water content is set equal to the field capacity parameter (i.e., the water holding capacity): [...]

If I understand it correctly, the  $\theta_i$  is initialized? I suggest to rename  $\Delta p$  to  $\Delta d_ice$ . Typically, *p* represent pressure. So, ice thickness is initialized too? Does the model start from the initialized ice thickness or there is a steady-state run? So, the second term in equation 2 should be less than or equal to 1? Otherwise subsidence could be greater than the ice thickness. Can that be the case? I did not understand the denominator. What is 1-phi\_{nat} mean?

The volumetric ice contents  $\theta_i$  are initialized as presented in Table 2. The excess ice content and the depth of the excess ice bearing layers can vary between different tiles (e.g. between polygon centres, rims and troughs). We did not conduct steady-state runs for the excess ice distribution, as our numerical model is not capable of simulating the accumulation of excess ice in the subsurface (see Discussion section 4.3). Instead, we based the cryostratigraphy on available measurements from the study area and previous modelling studies.

We renamed the variable  $\Delta p$  to  $\Delta d$ . Note, however, that it corresponds to the thickness of a soil cell which contains excess ice which is not to be confused with the thickness of the excess ice fraction. To clarify this point, and also to derive equation (2), we added a section in the Appendix (B), where we derive the formula and illustrate the composition of soil cells which contain excess ice (Figure B1). With these additional derivations it should be clear why the 1- $\phi_{nat}$  term is necessary, and that the  $\theta_x$  is indeed bounded between 0 and 1, so that the subsidence cannot be larger than the ice thickness.

**L149. Need a reference after "... hierarchical approach".**

This terminology was introduced in our article to refer to the novel approach to apply the tiling concept on multiple scales (micro- and meso-scale) at the same time. In the revised manuscript we dropped the adjective "hierarchical" as it might be misleading. Instead, we only speak of "multi-scale tiling". As far as we know, the concept has not been applied in this form to permafrost environments so that we do not have additional references to provide.

 $N^{\mu}$  is that somewhat standard notation? I had a hard time following that notation and remembering what it means. Is there a way to change it or use some other more intuitive notation? For example, use 1m2 or 1km2 notations. What is the total area modeled? Is this modeling represent a transect or a 2d area?

 $N^{\mu}$  and  $N^{m}$  refer to the number of tiles which have been used to represent the heterogeneities on the micro- and meso-scale, respectively. In our simulations, these are either set to 1 or 3, as mentioned in the listing of the model setups in Section 2.2.1 as well as in Figure 2. These variables do not directly relate to an area or a grid resolution. The tiling concept which we employed in this study does not use an explicit 2D or 3D mesh, but rather a combination of multiple (coupled) 1D "submodels", each of which is representative for a different landscape unit.

The questions of the reviewer indicate that the tile-based modelling approach was not sufficiently explained in the original article. We hence entirely revised the description of the tiling approach and added a schematic (Figure 1), which illustrates the concept in combination with the revised Figure 2.

Does homogeneous means that one tile represents the entire transect. If so, then it would be easier to say that 1 tile approach.

We renamed all model configurations for the revised manuscript and adopted the reviewer's suggestion to call the most simple setup "single-tile".

**What is the external reservoir? Is that water table depth?**

The external reservoir can be thought of as a constant water table exterior to the model domain. If a tile which is connected to such a reservoir has a water table which exceeds that of the external reservoir, excess water can run off into the reservoir (without changing the level of the reservoir).

**Figure 2, I had a hard time to understand and follow.**

We revised the Figure based on suggestions of reviewer #1 who otherwise found the Figure very helpful as it provides an overview of all setups. We are confident that the revised Figure layout, together with the extended description of the tile-based modelling approach, facilitate a more intuitive understanding of our methodology.

**L204 How many topological characteristics were used? Are these characteristics represent only magnitude of the lateral fluxes or something else too?**

We used the following geometrical relations to characterize the tiles: area of a tile, elevation of a tile, distance between adjacent tiles, contact length between adjacent tiles. These relations have been used to calculate the magnitude of lateral fluxes as explained in the preceding studies Nitzbon et al. (2019) and Nitzbon et al. (2020). In addition, the areal proportions of the tiles have been used to calculate the area-weighted mean thaw depth and accumulated subsidence which is displayed in Figures 7 and 8.

**Table 3 what is 'not a null' over 'big sigma' columns represent?**

The column shows the area-weighted mean or the sum of the different characteristics of the micro-scale tiles. For the revised manuscript, we split up the column into two and labeled it with text instead of symbols.

I like the results section and was able to make more sense of it. I think that discussing the geomorphological processes as well as figure 6 diversify the message: "the importance of the tile-approach adoption by the ESM type models." I guess, it is important to focus on that message instead of diving into the concepts and pathways of the polygonal tundra geomorphological evolution.

For this type of paper, I would like to see a more in-depth mathematical analysis of the difference between different spatial resolutions as well as discussion of the corresponding uncertainties. I understand that this might lead to way too much work and may not be feasible in this paper. Then I suggest to exclude the ESM modeling discussion from the article and give it a different angle from the beginning.

We are happy that the reviewer could follow the results section despite the lack of clarity in our method description. We agree with the reviewer that the adoption of our tile-based model-setup in coarse-scale model frameworks would deserve a more in-depth mathematical analysis and framing, which is, however, beyond the scope of our study. We still see several important implications of our work for the LSM/ESM community which we discuss in Section 4.2 of the revised manuscript.

In addition to this, we think that it is an important contribution of our work, that manifold pathways of landscape evolution can be simulated and that these involve various feedback processes which influence permafrost degradation in response to a warming climate. In the revised manuscript, we explain these (geomorphological) processes and feedbacks directly along with the results (following the suggestion of reviewer #1), and discuss them in Section 4.1. The extended discussion of the geomorphological evolution of ice-wedge polygons and thaw lakes in Section 4.3 is intended to highlight links between our modelling work and the efforts of field researchers to develop conceptual models of these processes and landforms.

**Consider bringing Figure 6 into the methods or introduction. Then it will setup the stage for the follow-up story.**

We thank the reviewer for the suggestion. However, we decided to keep this figure in the discussion section, as it illustrates not only the capacities of our modeling scheme but also its limitations.

**References (not contained in the original manuscript)**

Schneider von Deimling, T., Lee, H., Ingeman-Nielsen, T., Westermann, S., Romanovsky, V., Lamoureux, S., Walker, D. A., Chadburn, S., Cai, L., Trochim, E., Nitzbon, J., Jacobi, S., & Langer, M. (2020). Consequences of permafrost degradation for Arctic infrastructure – bridging the model gap between regional and engineering scales. The Cryosphere Discussions, 1–31. https://doi.org/10.5194/tc-2020-192